# A stable, distributed code for cue value in mouse cortex during reward learning

David J Ottenheimer[1,2,3†], Madelyn M Hjort[1,2†], Anna J Bowen[3†],
Nicholas A Steinmetz[1,3*‡], Garret D Stuber[1,2,4*‡]

[1]Center for the Neurobiology of Addiction, Pain and Emotion, University of Washington, Seattle, United States; [2]Anesthesiology and Pain Medicine, University of Washington, Seattle, United States; [3]Department of Biological Structure, University of Washington, Seattle, United States; [4]Department of Pharmacology, University of Washington, Seattle, United States

**\*For correspondence:**
nick.steinmetz@gmail.com
(NAS);
gstuber@uw.edu (GDS)

†These authors contributed
equally to this work
‡These authors also contributed
equally to this work

**Competing interest:** The authors
declare that no competing
interests exist.

Reviewing Editor: Michael B
Eisen, University of California,
Berkeley, United States

**Abstract** The ability to associate reward-predicting stimuli with adaptive behavior is frequently attributed to the prefrontal cortex, but the stimulus-specificity, spatial distribution, and stability of prefrontal cue-reward associations are unresolved. We trained head-fixed mice on an olfactory Pavlovian conditioning task and measured the coding properties of individual neurons across space (prefrontal, olfactory, and motor cortices) and time (multiple days). Neurons encoding cues or licks were most common in the olfactory and motor cortex, respectively. By quantifying the responses of cue-encoding neurons to six cues with varying probabilities of reward, we unexpectedly found value coding in all regions we sampled, with some enrichment in the prefrontal cortex. We further found that prefrontal cue and lick codes were preserved across days. Our results demonstrate that individual prefrontal neurons stably encode components of cue-reward learning within a larger spatial gradient of coding properties.

## eLife assessment

This study makes **valuable** observations about the representation of "value" in the mouse brain, by using a nice task design and recording from an impressive number of brain regions. The combination of state-of-the-art imaging and electrophysiology data offer **solid** support for the authors' conclusions. The paper will be of interest to a broad audience of neuroscientists interested in reward processing in the brain.

## Introduction

Association of environmental stimuli with rewards and the subsequent orchestration of value-guided reward-seeking behavior are crucial functions of the nervous system linked to the prefrontal cortex (PFC) (*Miller and Cohen, 2001*; *Klein-Flügge et al., 2022*). PFC is heterogeneous, with many studies noting subregional differences in both neural coding *Kennerley et al., 2009*; *Sul et al., 2010*; *Hunt et al., 2018*; *Wang et al., 2020a* and functional impact on *Dalley et al., 2004*; *Rudebeck et al., 2008*; *Buckley et al., 2009*; *Kesner and Churchwell, 2011* value-based reward seeking in primates and rodents. Furthermore, functional manipulations of PFC subregions exhibiting robust value signals do not always cause a discernible impact on reward-guided behavior (*Chudasama and Robbins, 2003*; *St Onge and Floresco, 2010*; *Dalton et al., 2016*; *Verharen et al., 2020*; *Wang et al., 2020a*), encouraging investigation of differences between value signals across PFC. Within individual PFC subregions, multiple studies have observed evolving neural representations across time, calling into question the stability of PFC signaling (*Hyman et al., 2012*; *Malagon-Vina et al., 2018*). A systematic

comparison of coding properties across rodent PFC and related motor and sensory regions, as well as across days and stimulus sets, is necessary to provide a full context for the contributions of PFC subregions to reward processing.

Identifying neural signals for value requires a number of considerations. One issue is that other task features can vary either meaningfully or spuriously with value. In particular, action coding is difficult to parse from value signaling, given the high correlations between behavior and task events (*Musall et al., 2019*; *Zagha et al., 2022*) and widespread neural coding of reward-seeking actions (*Steinmetz et al., 2019*). Additionally, without a sufficiently rich value axis, it is possible to misidentify neurons as 'value' coding even though they do not generalize to valuations in other contexts (*Stalnaker et al., 2015*; *Hayden and Niv, 2021*; *Zhou et al., 2021*). Because reports of the value have come from different experiments across different species, it is difficult to compare the presence of value signaling even across regions within the prefrontal cortex (*Kennerley et al., 2009*; *Sul et al., 2010*; *Stalnaker et al., 2015*; *Otis et al., 2017*; *Hunt et al., 2018*; *Namboodiri et al., 2019*; *Wang et al., 2020a*; *Hayden and Niv, 2021*; *Zhou et al., 2021*).

In this work, we sought to address the existing ambiguity in the distribution and stability of value signaling. We implemented an olfactory Pavlovian conditioning task that permitted the identification of value correlates within the domain of reward probability across two separate stimulus sets. With acute in vivo electrophysiology recordings, we were able to assess the coding of this task across 11 brain regions, including five PFC subregions, as well as olfactory and motor cortex, in a single group of mice, permitting a well-controlled comparison of coding patterns across a large group of the task-relevant regions in the same subjects. Unexpectedly, in contrast to the graded cue and lick coding across these regions, the proportion of neurons encoding cue value was more consistent across regions, with a slight enrichment in PFC but with similar value decoding performance across all regions. To assess coding stability, we performed 2-photon calcium imaging of neurons in the PFC for multiple days and determined that the cue and lick codes we identified were stable over time. Our data demonstrate the universality and stability of cue-reward coding in the mouse cortex.

## Results
### Distributed neural activity during an olfactory Pavlovian conditioning task

We trained mice on an olfactory Pavlovian conditioning task with three cue (conditioned stimulus) types that predicted reward on 100% ('CS+'), 50% ('CS50'), or 0% ('CS−') of trials (*Figure 1A*). Each mouse learned two odor sets (odor sets A and B), trained and imaged on separate days and then, for electrophysiology experiments, presented in six alternating blocks of 51 trials during the recording sessions (*Figure 1B*). Mice developed anticipatory licking (*Figure 1C–D*), and the rate of this licking correlated with reward probability (*Figure 1—figure supplement 1*), indicating that subjects successfully learned the meaning of all six odors.

Using Neuropixels 1.0 and 2.0 probes (*Jun et al., 2017*; *Steinmetz et al., 2021*), we recorded the activity of individual neurons in PFC, including anterior cingulate area (ACA), frontal pole (FRP), prelimbic area (PL), infralimbic area (ILA), and orbital area (ORB) (*Wang et al., 2020b*; *Laubach et al., 2018*). We also recorded from: secondary motor cortex (MOs), including anterolateral motor cotex (ALM), which has a well-characterized role in licking *Chen et al., 2017*; olfactory cortex (OLF), including dorsal peduncular area (DP), dorsal taenia tecta (TTd), and anterior olfactory nucleus (AON), which receive input from the olfactory bulb (*Igarashi et al., 2012*; *Mori and Sakano, 2021*); and striatum, including caudoputamen (CP) and nucleus accumbens (ACB), which are major outputs of PFC (*Heilbronner et al., 2016*, *Figure 1E–F*). In a separate group of mice, we performed longitudinal 2-photon calcium imaging through a Gradient Refractive Index (GRIN) lens to track the activity of individual neurons in PL across several days of behavioral training (*Figure 1G–H*). Both techniques permitted robust measurement of the activity of neurons of interest and generated complementary results (*Figure 1I–J*, *Figure 1—figure supplement 2*).

### Graded cue and lick coding across the recorded regions

In the electrophysiology experiment, we isolated the spiking activity of 5332 individual neurons in regions of interest across 5 mice (449-1550 neurons per mouse, *Figure 2A*, *Figure 2—figure*

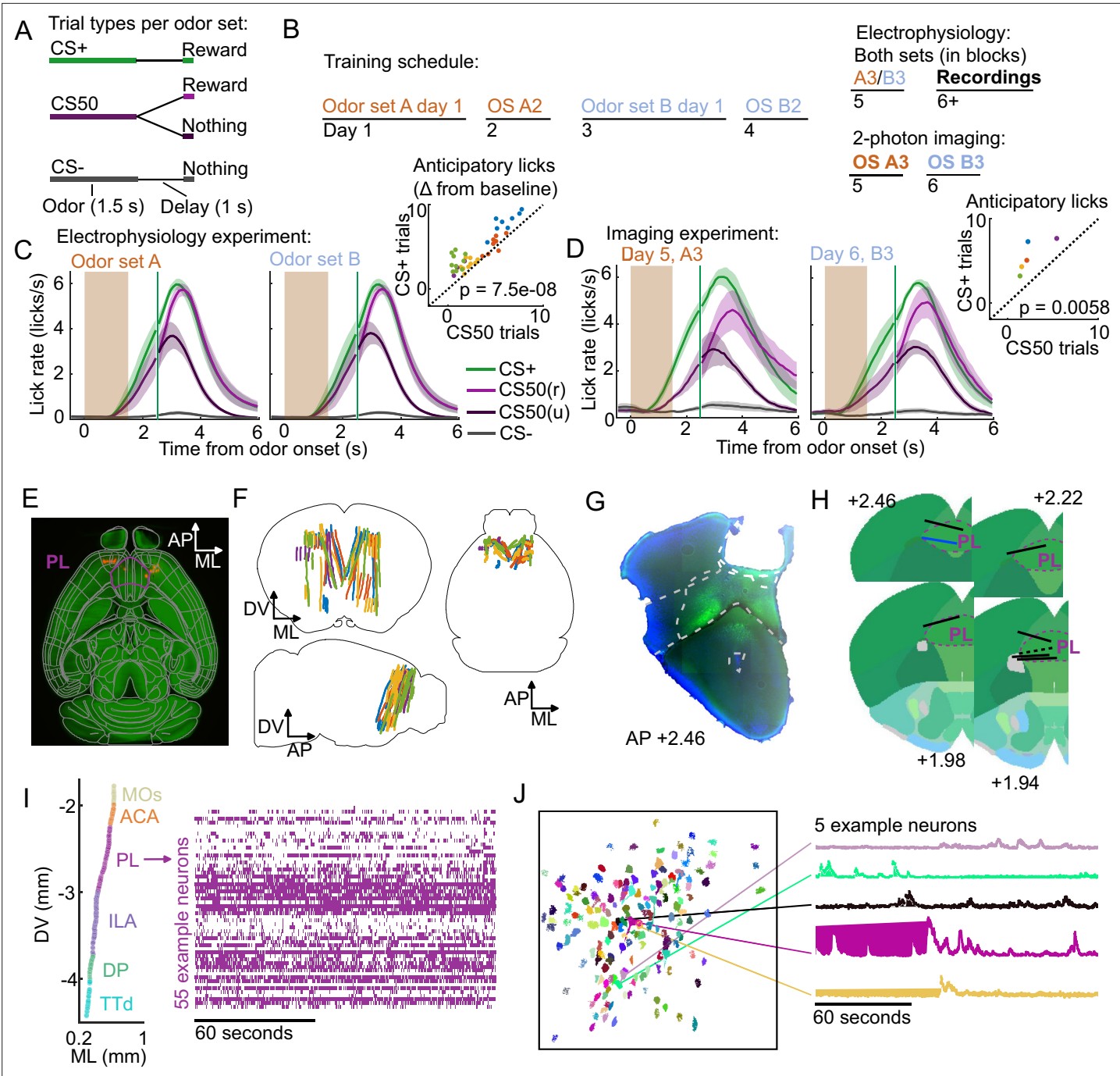

**Figure 1.** Electrophysiology and calcium imaging during olfactory Pavlovian conditioning. (**A**) Trial structure in Pavlovian conditioning task. (**B**) Timeline for mouse training. (**C**) Mean (+/− standard error of the mean (SEM)) lick rate across mice ($n = 5$) on each trial type for each odor set during electrophysiology sessions. CS50(r) and CS50(u) are rewarded and unrewarded trials, respectively. Inset: mean anticipatory licks (change from baseline) for the CS+ and CS50 cues for every session, color-coded by mouse. $F_{(1,66)} = 36.6$ for a main effect of cue in a two-way ANOVA including an effect of subject. (**D**) Same as (**C**), for the third session of each odor set ($n = 5$ mice). $t_{(4)} = -5.4$ for a t-test comparing anticipatory licks on CS+ and CS50 trials. (**E**) Neuropixels probe tracks labeled with fluorescent dye (red) in cleared brain (autofluorescence, green). AP, anterior/posterior; ML, medial/lateral; DV, dorsal/ventral. Allen common-coordinate framework (CCF) regions delineated in gray. Outline of prelimbic area in purple (**F**) Reconstructed recording sites from all tracked probe insertions ($n = 44$ insertions, $n = 5$ mice), colored by mouse. (**G**) Sample histology image of lens placement. Visualization includes DAPI (blue) and GCaMP (green) signal with lines indicating cortical regions from Allen Mouse Brain Common Coordinate Framework. (**H**) Location of all lenses from experimental animals registered to Allen Mouse Brain Common Coordinate Framework. Blue line indicates location of lens in (**A**). The dotted black line represents approximate location of tissue that was too damaged to reconstruct an accurate lens track. The white

*Figure 1 continued on next page*

*Figure 1 continued*

dotted line indicates prelimbic area (PL) borders.(**I**) ML and DV coordinates of all neurons recorded in one example session, colored by region, and spike raster from example PL neurons. (**J**) ROI masks for identified neurons and fluorescence traces from five example neurons.

The online version of this article includes the following figure supplement(s) for figure 1:

**Figure supplement 1.** Anticipatory licking during the electrophysiology sessions.

**Figure supplement 2.** Similar neural activity in prelimbic area using electrophysiology and calcium imaging.

*supplement 1A*). The activity of neurons in all regions exhibited varying degrees of modulation in response to the six cues (*Figure 2B*). Broadly, there was strong modulation on CS+ and CS50 trials that appeared to be common to both odor sets (*Figure 2—figure supplement 1B*). Across regions, there was heterogeneity in both the magnitude and the timing of the neural modulation relative to odor onset (*Figure 2—figure supplement 1C*).

To quantify the relative contribution of cues and conditioned responding (licking) to the activity of neurons in each region, we implemented reduced rank kernel regression (*Steinmetz et al., 2019*), using cues, licks, and rewards to predict neurons' activity on held-out trials (*Figure 2C*, *Figure 2—figure supplement 2A*). To determine the contribution of cues, licks, and rewards to each neuron's activity, we calculated unique variance explained by individually removing each predictor from the model and calculating the reduction in model performance (*Figure 2D*).

We identified individual neurons encoding cues, licks, or rewards as those for which that predictor uniquely contributed to 2% or more of their variance (a cutoff permitting no false positives and identifying neurons with robust task modulation, see Methods and *Figure 2—figure supplement 3*). Neurons encoding cues (24% of all neurons), licks (11%), or both (16%) were most common. Neurons with any response to reward (independent of licking) were rare (5%) (*Horst and Laubach, 2013*). Cue neurons were characterized by sharp responses aligned to odor onset; in contrast, lick neurons' responses were delayed and peaked around reward delivery (*Figure 2—figure supplement 2B–C*), consistent with the timing of licks (*Figure 1C*). The activity of cue neurons on rewarded and unrewarded CS50 trials validated our successful isolation of neurons with cue but not lick responses (*Figure 2—figure supplement 2D*). The spatial distributions of cue and lick cells were noticeably different (*Figure 2E*). The differences could be described as graded across regions, with the most lick neurons in ALM, and the most cue neurons in olfactory cortex and ORB, though each type of neuron was observed in every region (*Figure 2F–G*, *Figure 2—figure supplement 4*). Thus, our quantification of task encoding revealed varying proportions of cue and lick signaling across all regions.

## Cue value coding is present in all regions

To expand upon our analysis identifying cue-responsive neurons, we next assessed the presence of cue value coding in this population. The three cue types (CS+, CS50, or CS−) in our behavioral tasks varied in relative value according to the predicted probability of reward (*Fiorillo et al., 2003*; *Eshel et al., 2016*; *Winkelmeier et al., 2022*). We reasoned that a neuron encoding cue value should have activities that scaled with the relative value of the cues (*Figure 3A*). We modeled this relationship on a per-neuron basis by scaling a single cue kernel by its reward probability (0, 0.5, or 1, see Methods, *Figure 3B*). This model describes cue activity as similar across odors of the same value, and scaling in magnitude according to each odor's value. To consider alternative cue coding patterns, we also fit each neuron with 152 additional models containing all possible permutations of these values across the six cues, as well as models with selective responses for 1, 2, 3, 4, 5, or 6 cues, and determined which model best fit each neuron (*Figure 3—figure supplement 1*). If cue responses were exclusively sensory and followed known olfactory coding properties (*Stettler and Axel, 2009*; *Pashkovski et al., 2020*), there would be no bias toward the ranked value model (CS+>CS50>CS−). We found, however, that this model was the most frequent best model, accounting for 14% of cue neurons (*Figure 3C*). We refer to these neurons as value cells. There were two additional patterns that emerged across the population of cue neurons. First, there was a large fraction best explained by the model with equivalent responses to all 6 cues, which we term untuned cells (14% of cue neurons). Second, many of the alternative models had coding patterns that were similar to the ranked value model, and these appeared to be overrepresented among cue neurons, as well. We quantified the similarity to ranked value by correlating the values assigned to each cue in each model with those assigned to the cues

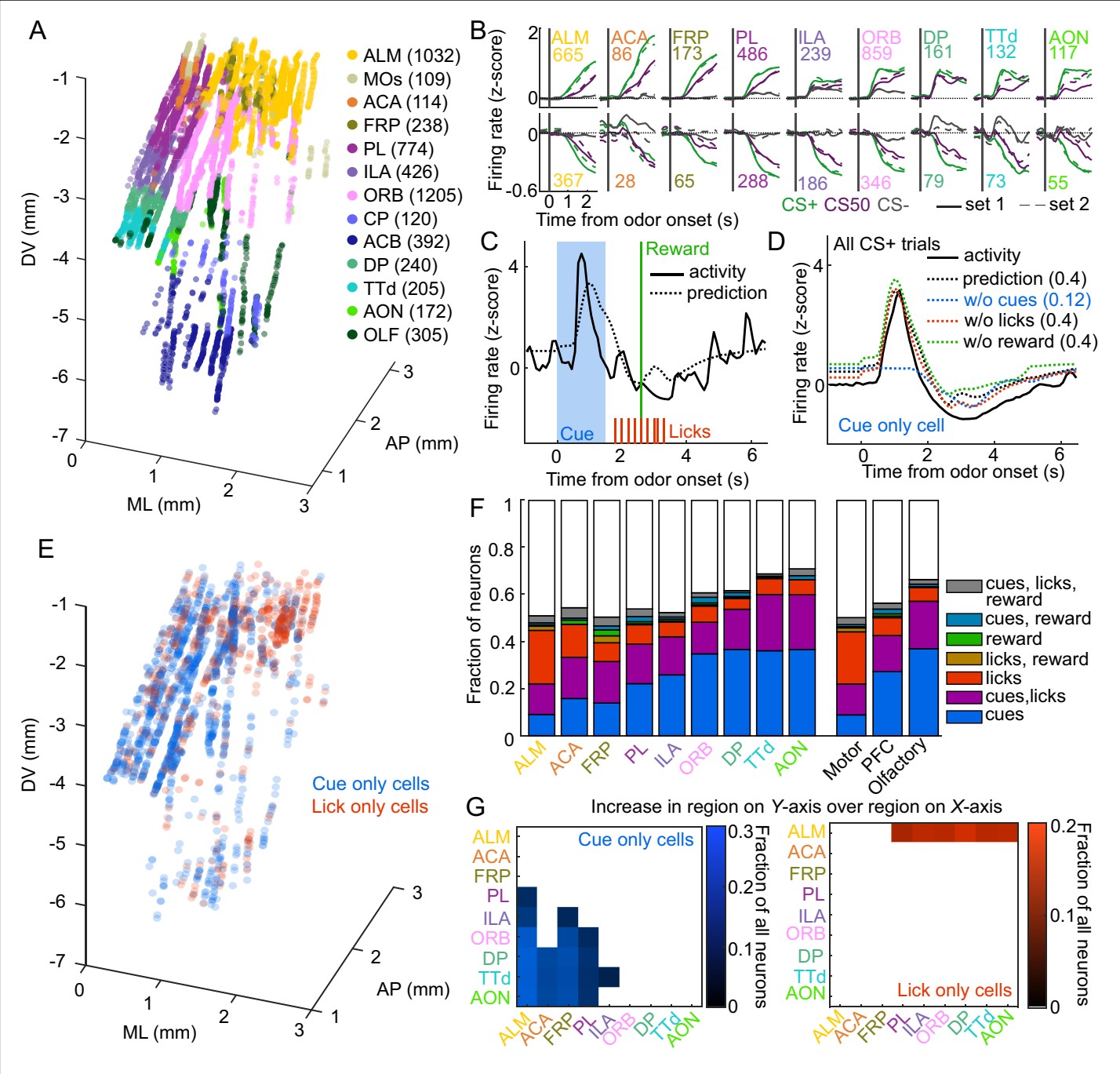

**Figure 2.** Graded cue and lick coding across the recorded regions. (**A**) Location of each recorded neuron relative to bregma, projected onto one hemisphere. Each neuron is colored by common-coordinate framework (CCF) region. Numbers indicate total neurons passing quality control from each region. (**B**) Mean normalized activity of all neurons from each region, aligned to odor onset, grouped by whether peak cue activity (0–2.5 s) was above (top) or below (bottom) baseline in held out trials. Number of neurons noted for each plot. (**C**) Example kernel regression prediction of an individual neuron's normalized activity on an example trial. (**D**) CS+ trial activity from an example neuron and predictions with full model and with cues, licks, and reward removed. Numbers in parentheses are model performance (fraction of variance explained). (**E**) Coordinates relative to bregma of every neuron encoding only cues or only licks, projected onto one hemisphere. (**F**) Fraction of neurons in each region and region group classified as coding cues, licks, reward, or all combinations of the three. (**G**) Additional cue (left) or lick (right) neurons in region on Y-axis compared to region on x-axis as a fraction of all neurons, for regions with statistically different proportions (see Methods).

The online version of this article includes the following figure supplement(s) for figure 2:

**Figure supplement 1.** Task-related neural activity across brain regions.

*Figure 2 continued on next page*

*Figure 2 continued*

**Figure supplement 2.** Identification of cue and lick cells with GLM.

**Figure supplement 3.** Validation of variance cutoff for variable coding.

**Figure supplement 4.** Comparing proportions of cue and lick neurons across regions.

in the ranked value model; this approach revealed an enrichment in neurons best fit by models most similar to ranked value (35% of cue neurons, *Figure 3C–D*). We refer to neurons best fit by models most similar to the value model as value-like cells.

Value characteristics were particularly strong among the neurons we identified as value cells. In particular, there was strong modulation for the CS+ odors, moderate modulation for CS50 odors, and the least modulation for CS− odors (*Figure 3E*). These characteristics were present to varying degrees in value-like cells, as well (*Figure 3F*). A key characteristic of value cells, however, was the singular value axis on which the cues were encoded. This was evident when projecting population activity onto the dimensions separating CS+ trials from CS− trials and CS50 trials from CS− trials (*Figure 3—figure supplement 2A*); the trajectory of value neurons traveled the same angle in this space for CS+ and CS50 trials, but differed for value-like (*Figure 3—figure supplement 2B*). We additionally characterized the coding properties of these populations with single-unit and pseudo ensemble decoding. For individual neurons decoding the six cue identities, performance was better using value cells than value-like or untuned cells (*Figure 3G*). At the population level, however, all groups of neurons performed similarly (*Figure 3H*). A key feature of a value signal beyond decoding cue identity, though, is the ability to represent many distinct cues along a shared value axis. Therefore, the value cells should be able to decode the value of a cue never presented during the training of the model. With this approach, models trained on value cells had better predictions of held-out cue value, leading to higher decoding accuracy (CS+, CS50, or CS−), compared to value-like and untuned cells (*Figure 3I*). Therefore, we successfully identified a population of neurons strongly encoding key features of value.

Interestingly, the frequency of value cells was similar across the recorded regions (*Figure 4A*). Despite the regional variability in the number of cue cells broadly (*Figure 2F–G*), there were very few regions that statistically differed in their proportions of value cells (*Figure 4A*, *Figure 4—figure supplement 1*). Overall, there were slightly more value cells across all of PFC than in motor and olfactory cortex (*Figure 4A*, *Figure 4—figure supplement 1*). Although the olfactory cortex had the most cue cells, these were less likely to encode value than cue cells in other regions (*Figure 4—figure supplement 2*). Value-like cells were also widespread; they were less frequent in the motor cortex as a fraction of all neurons, but they were equivalently distributed in all regions as a fraction of cue neurons (*Figure 4B*, *Figure 4—figure supplement 1*, *Figure 4—figure supplement 2*).

We next investigated the robustness of the value representation in each of our recorded regions. Principal component analysis on value and value-like cells from each region revealed similarly strong value-related dynamics across motor, prefrontal, and olfactory regions (*Figure 4C–D*). We quantified the robustness of value coding in each region by decoding cue value using selections of value cells from each region and found similar performance across all regions (*Figure 1E*). Taken together, these data illustrate that, in contrast to cue and lick coding broadly, value coding is similarly represented across the regions we sampled. In fact, this observation extended to the striatal regions we sampled as well, indicating that such value coding is widespread even beyond cortex (*Figure 4—figure supplement 3*).

Because cue valuations can be influenced by preceding reward outcomes, we next considered whether the cue value signaling we detected was sensitive to the history of reinforcement (*Nakahara et al., 2004*; *Ottenheimer et al., 2020*; *Winkelmeier et al., 2022*). To estimate the subjects' trial-by-trial cue valuation, we fit a linear model predicting the number of anticipatory licks on each trial using cue type, overall reward history, and cue type-specific reward history as predictors. We found a strong influence of cue type-specific reward history and a more modest influence of overall reward history (*Figure 5A*). We used the model prediction of licks per trial as our estimate of trial value; the effects of reward history on lick rate were apparent when grouping trials by the value estimates from the trial value model (*Figure 5B*).

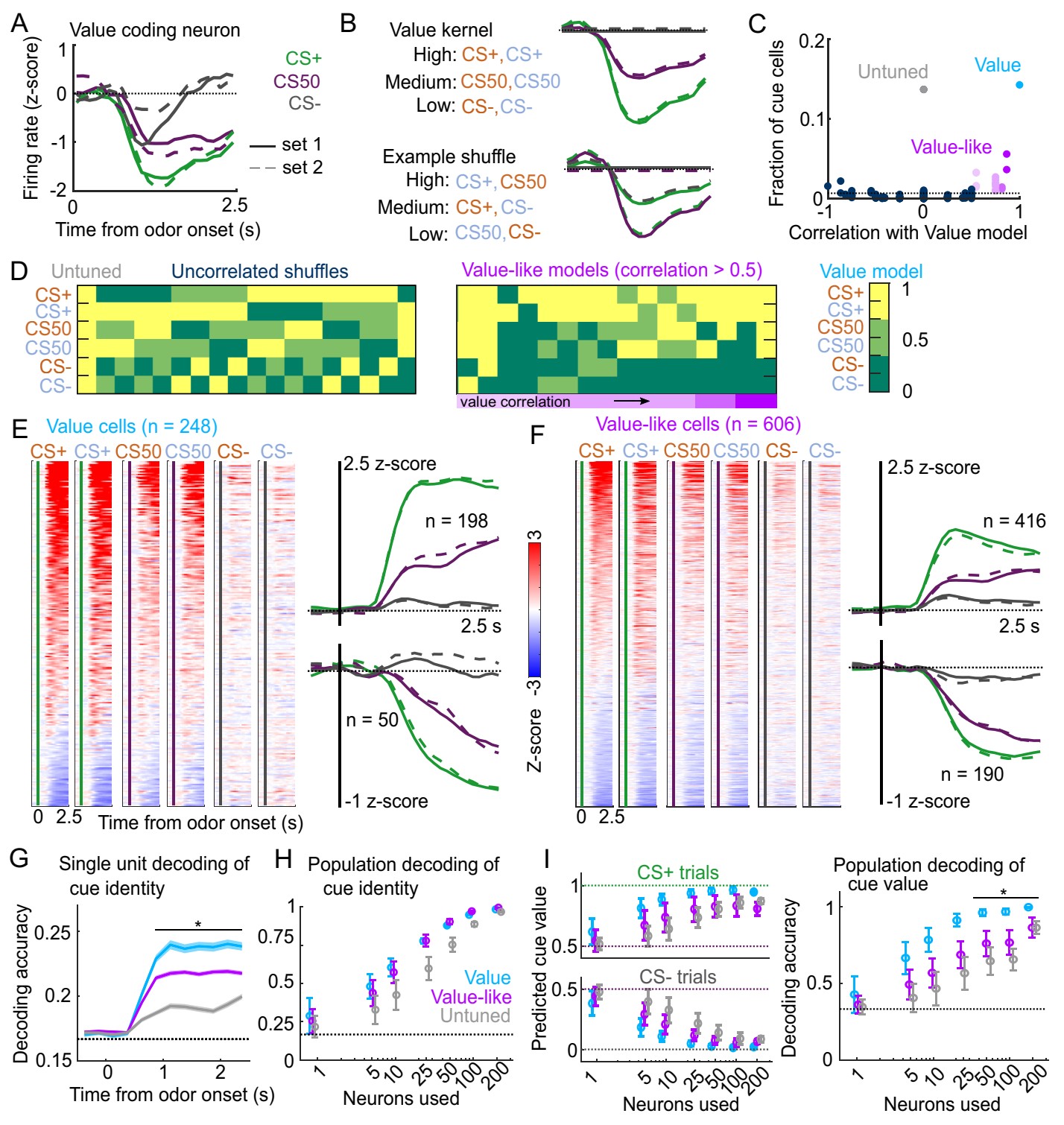

**Figure 3.** Robust value encoding and decoding among cue cells. (**A**) Normalized activity of an example value cell with increasing modulation for cues with higher reward probability.(**B**) For the same neuron, model-fit cue kernel for the original value model and with one of the 152 alternatively-permuted cue coding models. (**C**) Distribution of best model fits across all cue neurons. Light blue is value model, purple is value-like models, gray is untuned model, and the remaining models are dark blue. Value-like models are shaded according to their correlation with ranked value, as illustrated in (**D**). Dashed line is chance proportion when assuming even distribution. (**D**) Schematic of value assigned to each of the six cues for many of the cue coding models (full schematic in **Figure 3—figure supplement 1**). Value-like models are sorted by their correlation with the ranked value model. (**E**) Left: normalized activity of every value cell, sorted by mean firing 0–1.5s following odor set A CS+ onset. Right: mean normalized activity of all value

*Figure 3 continued on next page*

*Figure 3 continued*

cells, grouped by whether peak cue activity (0–2.5s) was above (top) or below (bottom) baseline in held out trials. Number of neurons noted for each plot. (**F**) As in (**E**), for value-like cells. (**G**) Accuracy (mean ± SEM across neurons) of decoded cue identity for single neurons of value ($n = 248$), value-like ($n = 606$), and untuned ($n = 238$) neurons. * indicates where value, value-like, and untuned neurons significantly differed from each other and baseline (all $p < 0.001$, Bonferroni corrected). All pairwise comparisons in *Supplementary file 2*. (**H**) Accuracy (mean ± SD across bootstrapped iterations) of decoded cue identity using different numbers of neurons. (**I**) Left: estimated value (mean ± SD across 1000 bootstrapped iterations) of held out CS+ (top) and CS− (bottom) trials using linear models trained on the activity of value, value-like, or untuned neurons. Right: accuracy (mean ± SD across bootstrapped iterations) of decoded cue value using these value estimates. * indicates where the accuracy of value neurons exceeded value-like and untuned neurons (all $p < 0.016$, bootstrapped). All pairwise comparisons in *Supplementary file 2*.

The online version of this article includes the following figure supplement(s) for figure 3:

**Figure supplement 1.** Schematic of value model shuffles.

**Figure supplement 2.** Population analysis of value coding schemes.

We, therefore, investigated whether value cells showed similar trial-by-trial differences in their cue-evoked firing rates (*Figure 5C*). To test this, we compared the fit of our original cue coding models (*Figure 3B–D*) with an alternative model in which the kernel scaled with the per-trial value estimates from our trial value model (*Figure 5D*). Overall, 5% of cue cells, including 15% of the value cells, were best fit by the history model. Although the number of anticipatory licks per trial was used to generate the trial value estimates, the precise licking pattern on those trials was a poorer predictor of neural responses than the trial value-scaled cue kernel model (*Figure 5E*). To further evaluate the history component of these neurons, we calculated these neurons' activity on CS50 trials of varying value estimates from the trial value model and projected it onto the population dimension maximizing the separation between CS+ and CS−. We hypothesized that high value CS50 trials would be closer to CS+ activity while low value CS50 trials would be closer to CS− activity. Indeed, history cells (and lick cells) demonstrated graded activity along this dimension, in contrast to non-history value, value-like, and untuned cells (*Figure 5F–H*). Finally, we examined the regional distribution of history cells and found low numbers across all regions, but with a higher prevalence overall in PFC than in motor and olfactory cortex (*Figure 5I*), lending additional support for slightly enhanced value coding in PFC.

## Cue coding emerges along with behavioral learning

To determine the timescales over which these coding schemes emerged and persisted, we performed longitudinal 2-photon calcium imaging and tracked the activity of individual neurons across several days of behavioral training (*Figure 6A*). We targeted a GRIN lens to PL, a location with robust cue and lick coding (*Figure 2F*) and where cue responses were predominantly value or value-like (*Figure 4A–B*, *Figure 4—figure supplement 2*). Mice ($n = 8$) developed anticipatory licking during the first sessions of odor set A (A1) that differentiated CS+ trials from CS50 ($t(7) = 3.2$, $p = 0.015$) and CS− ($t(7) = 7.0$, $p = 0.0002$) trials and CS50 trials from CS− ($t(7) = 3.7$, $p = 0.008$) trials (*Figure 6B–C*). Visualizing the normalized activity across the imaging plane following CS+ presentation early and late in session A1 revealed a pronounced increase in modulation across this first session (*Figure 6D–E*). Individual neurons ($n = 705$, 41-165 per mouse) also displayed a notable increase in modulation in response to the CS+ after task learning (*Figure 6F*).

To determine whether this increase in activity was best explained by a cue-evoked response, licking, or both, we again used kernel regression to fit and predict the activity of each neuron for early, middle, and late trials in session A1. The number of individual neurons encoding cues more than doubled from early to late A1 trials (*Figure 6G*). The unique variance cues increased across this first session, in contrast to licks and reward (*Figure 6H*). This stark change in cue coding was also noticeable when plotting neurons encoding cues, licks, or both, as defined at the end of the sessions, on both early and late trials (*Figure 6I*). These data indicated that PFC neural activity related to cues (but not licks) rapidly emerge during initial learning of the behavioral task.

## Cue and lick coding is stable across days

We next assessed whether cue and lick coding were stable across days. By revisiting the same imaging plane on each day of training, we were able to identify neurons that were present on all three days of odor set A training ($n = 371$, 20-65 per mouse) (*Figure 7A–B*). There was remarkable conservation of task responding across days, both on an individual neuron level (*Figure 7C*) and across all

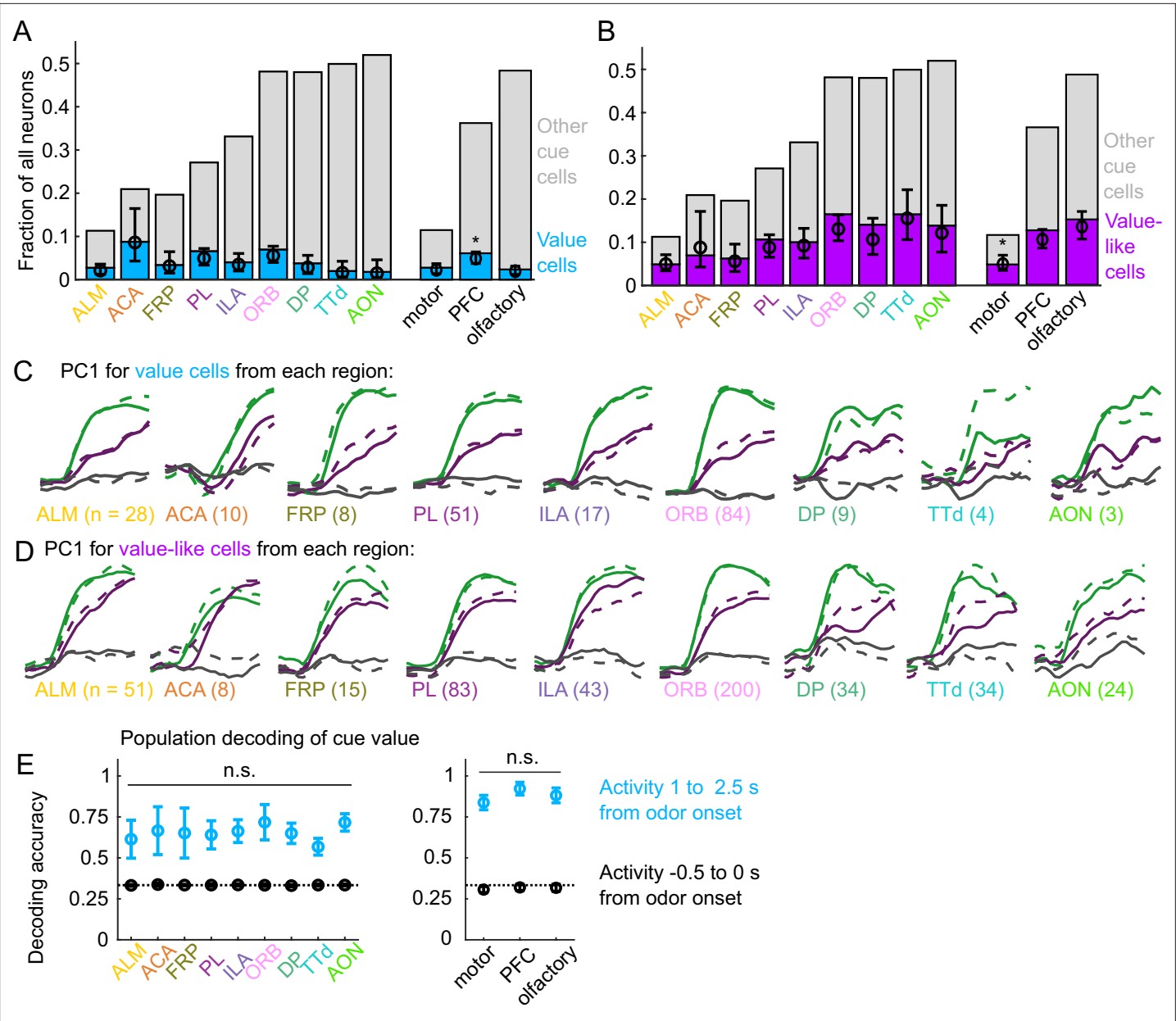

**Figure 4.** Widespread cue value coding. (**A**) Fraction of neurons in each region and region group classified as value cells (blue) and other cue neurons (gray), as well as fraction (± 95% CI) estimated from a linear mixed effects model with random effect of session (see Methods). Prefrontal cortex (PFC) has more value cells than motor ($p = 0.002$) and olfactory ($p = 0.00005$) cortex. All pairwise comparisons in *Supplementary file 3*. (**B**) As in (**A**), for value-like cells. Motor cortex has fewer value-like cells than PFC ($p = 8 * 10^{-6}$) and olfactory cortex ($p = 4 * 10^{-8}$). All pairwise comparisons in *Supplementary file 3*. (**C**) First principal component value cells from all regions. (**D**) As in (**C**), for value-like cells. (**E**) Accuracy of decoded cue value (mean ± SD across 1000 bootstrapped iterations) as in *Figure 3I*, using five (with replacement) value cells from each region (left) and 25 value cells from each region group (right) using cue-evoked (blue) and baseline (black) activity. No regions or region groups significantly differed from each other ($p > 0.46$, Bonferroni corrected). All pairwise comparisons in *Supplementary file 3*.

The online version of this article includes the following figure supplement(s) for figure 4:

**Figure supplement 1.** Relative proportions of value and value-like cells across regions.

**Figure supplement 2.** Value coding as a proportion of cue cells.

**Figure supplement 3.** Comparing prefrontal cortex (PFC) and striatum.

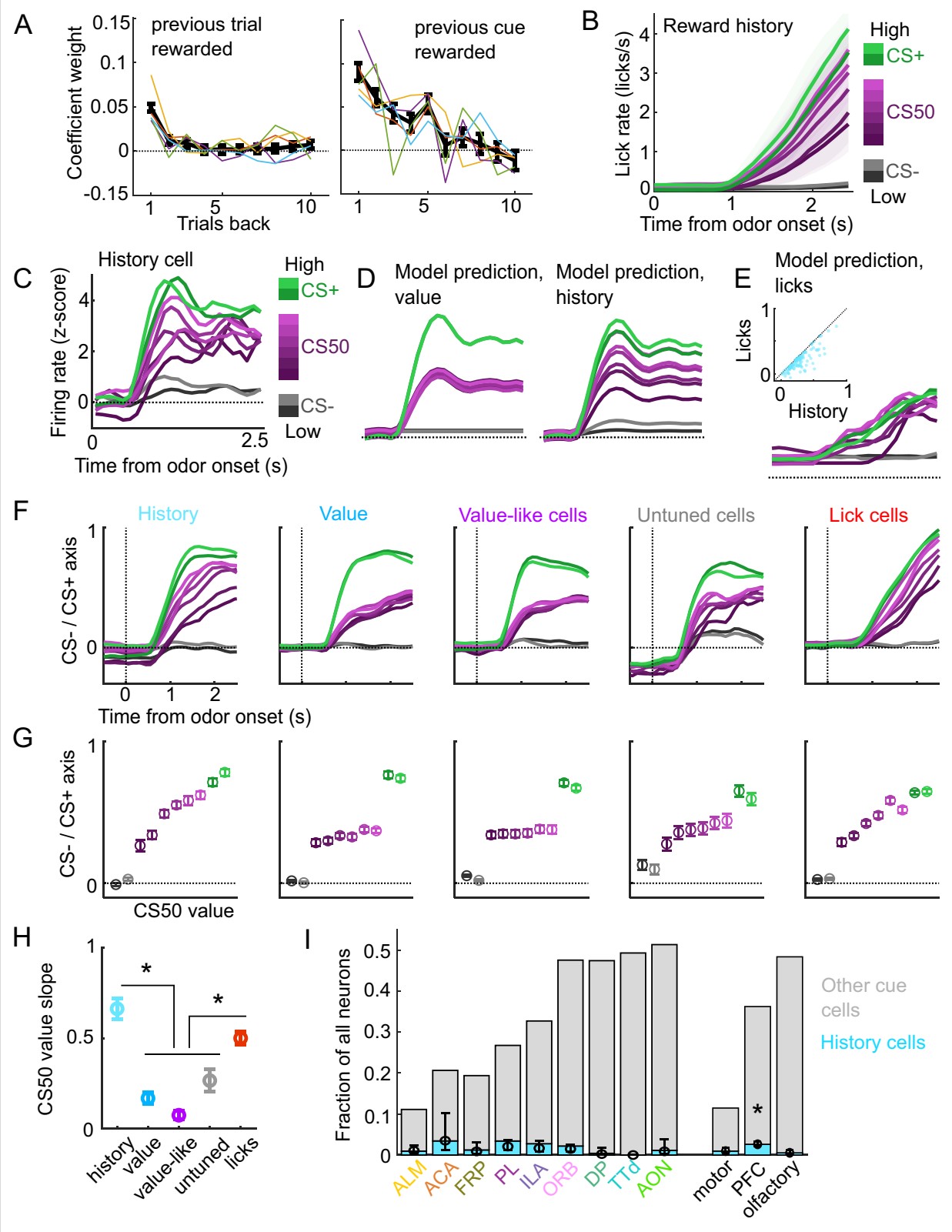

**Figure 5.** A subset of cue cells incorporate reward history. (**A**) Coefficient weight (± standard error from model fit) for reward outcome on the previous 10 trials of any type (left) and on the previous 10 trials of the same cue type (right) for the 'trial value' model: a linear model predicting the number of anticipatory licks on every trial of every session. Lick rates were normalized so that the maximum lick rate for each session was equal to 1. Colored lines are models fit to each individual mouse. (**B**) Mean (± SEM) lick rate across mice ($n = 5$ mice) on trials binned according to value estimated from

*Figure 5 continued on next page*

*Figure 5 continued*

the trial value model. (**C**) Normalized activity of an example history value cell with increasing modulation for cues of higher value. (**D**) For the same neuron, model-predicted activity with the original value model (left) and with the history model, which uses trial-by-trial value estimates from the trial value model (right). (**E**) For the same neuron, model-predicted activity using licks. Inset: variance explained using licks versus history for history neurons. (**F**) The activity of all cells in each category projected onto the coding dimension maximally separating CS− and CS+ for trials binned by value estimated from the trial value model. (**G**) The mean (± SD across 5000 bootstrapped selections of neurons) activity (1–2.5s from odor onset) along the coding dimension maximally separating CS− and CS+ for trials binned by value estimated from the lick model. (**H**) The mean (± SD across 5000 bootstrapped selections of neurons) slope of the activity on CS50 trials regressed onto the trial value model estimate for those trials. History and lick cells had greater slopes than the other groups ($p < 0.0003$, see ***Supplementary file 4***). (**I**) Fraction of neurons in each region and region group classified as history cells (light blue) and other cue neurons (gray), as well as estimated fraction (± 95% CI) with random effect of session (see Methods). Prefrontal cortex (PFC) had more history cells than motor ($p = 0.0016$) and olfactory ($p = 0.00053$) cortex. All pairwise comparisons in ***Supplementary file 4***.

imaged neurons (***Figure 7D***). In fact, neurons were much more correlated with their own activity on the subsequent day than would be expected by chance (***Figure 7E***, ***Figure 7—figure supplement 1A***). To further quantify coding stability, we fit our kernel regression to the activity of each neurons on session A3 (***Figure 7F***) and then used these models to predict activity in early, middle, and late trials on sessions A1-3. Session A3 model predictions were most highly correlated with true activity during A3, but they outperformed shuffle controls at all time points, demonstrating preservation of a learned coding scheme (***Figure 7G***, ***Figure 7—figure supplement 1B***). We then asked more specifically whether cells coding cues, licks, and both maintained their coding preferences across days. For each group of cells, we calculated their unique cue, lick, and reward variance at each time point. The preferred coding of each group, as defined in session A3, was preserved in earlier days (***Figure 7H***). Thus, cue and lick coding are stable properties of PFC neurons across multiple days of behavioral training.

A subset of mice ($n = 5$) also learned a second odor set (odor set B), presented on separate days. Activity was very similar for both odor sets, evident across the entire imaging plane (***Figure 8A***), for individual tracked neurons ($n = 594$, 81-153 per mouse) (***Figure 1—figure supplement 2B***), and for kernel regression classification of these neurons (***Figure 8B***). Notably, odor set A models performed similarly well at predicting both odors set A and odor set B activity (***Figure 8C***). Moreover, cue, lick, and both neurons maintained their unique variance preference across odor sets (***Figure 8D***). Finally, to investigate the presence of value coding across odor sets over separate days, we fit tracked cue neurons with the value model and its shuffles. Even with odor sets imaged on separate days (days 5 and 6 of training, A3 and B3), we again found that the value and value-like models were the best models for sizable fractions (9% and 47%, respectively) of cue neurons, demonstrating that value coding is conserved across stimulus sets on consecutive days (***Figure 8E–G***). Given the prominence of value-like signals in this imaged population, we then assessed the stability of cue cells with preferential CS+ responses across the tracked A1-3 sessions and found conservation of a value-like coding pattern (***Figure 8H***) and, as with the whole population (***Figure 7G***), greater correlation in activity across days than expected by chance (***Figure 8I***).

## Discussion

Our experiments assessed how coding for reward-predicting cues and reward-seeking actions differed across brain regions and across multiple days of training. We found coding for cues and licks in all regions we sampled, but their proportions varied in a graded way across those regions. In contrast to regional differences in the proportion of cue-responsive neurons, cue-value cells were present in all regions and value could be decoded from them with similar accuracy regardless of the region. Coding for cue value was greatly overrepresented compared to alternative cue coding schemes and, in a subset of neurons, incorporated the recent reward history. Cue coding was established within the first day of training and neurons encoding cues or licks maintained their coding preference across multiple days of the task; the value characteristics of cue cells were also maintained across days. These results demonstrate widespread value coding and stability of cue and lick codes in PFC.

### Graded cue and lick coding across regions

We found robust and separable coding for licks and cues (and combined coding of both) in all regions using electrophysiology and in PL using calcium imaging. The widespread presence of lick coding

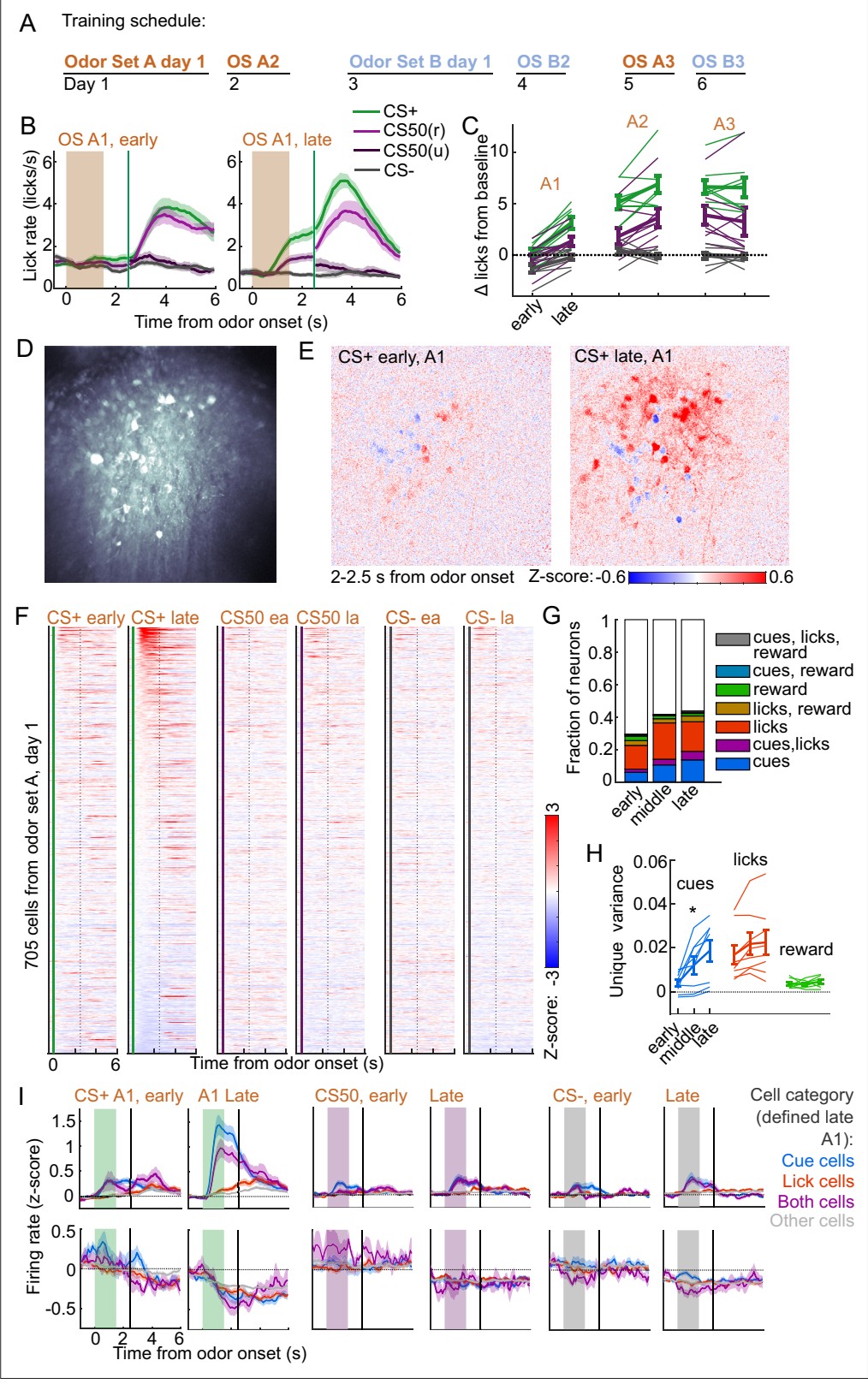

**Figure 6.** Acquisition of conditioned behavior and cue encoding in prefrontal cortex (PFC). (**A**) Training schedule for five of the mice in the calcium imaging experiment. An additional three were trained only on odor set A. (**B**) Mean (± SEM) licking on early (first 60) and late (last 60) trials from day 1 of odor set A ($n = 8$ mice). (**C**) Mean (± SEM) baseline-subtracted anticipatory licks for early and late trials from each day of odor set A. Thin lines are

*Figure 6 continued on next page*

*Figure 6 continued*

individual mice ($n = 8$ mice). (**D**) Standard deviation of fluorescence from example imaging plane. (**E**) Normalized activity of each pixel following CS+ presentation on early and late trials of session A1. (**F**) Normalized deconvolved spike rate of all individual neurons on early and late trials of session A1. (**G**) Proportion of neurons classified as coding cues, licks, rewards, and all combinations for each third of session A1. (**H**) Mean(± SEM across mice) unique variance explained by cues, licks, and rewards for neurons from each mouse. Thin lines are individual mice. Unique variance was significantly different across session thirds for cues ($F(2, 21) = 3.71$, $p = 0.04$) but not licks ($F(2, 21) = 0.37$, $p = 0.69$) or reward ($F(2, 21) = 0.65$, $p = 0.53$, $n = 8$ mice, one-way ANOVA). (**I**) Mean (± SEM) normalized deconvolved spike rate for cells coding cues ($n = 84$ above, $n = 28$ below), licks ($n = 91$ above, $n = 40$ below), both ($n = 31$ above, $n = 9$ below), or neither ($n = 307$ above, $n = 153$ below) on early and late trials, sorted by whether peak cue activity (0–2.5 s) was above (top) or below (bottom) baseline for late trials.

is consistent with recent reports of distributed movement and action coding (*Stringer et al., 2019*; *Musall et al., 2019*; *Steinmetz et al., 2019*); however, we saw sizable differences in the amount of lick coding across recorded regions. Notably, ALM had the greatest number of lick neurons, as well as the fewest cue neurons, perhaps reflecting its specialized role in the preparation and execution of licking behavior (*Chen et al., 2017*). Conversely, the olfactory cortical regions DP, TTd, and AON had the most cue neurons (especially non-value coding cue neurons), suggesting a role in early odor identification and processing (*Mori and Sakano, 2021*). PFC subregions balanced lick and cue coding, consistent with their proposed roles as association areas (*Miller and Cohen, 2001*; *Klein-Flügge et al., 2022*), but there was variability within PFC as well. In particular, ORB had a greater fraction of cue cells than any other subregions, consistent with its known dense inputs from the olfactory system (*Price, 1985*; *Price et al., 1991*; *Ekstrand et al., 2001*). Thus, our results establish that the neural correlates of this Pavlovian conditioned behavior consist of a gradient of cue and response coding rather than segmentation of sensory and motor responses.

## Widespread value signaling

Value signals can take on many forms and occur throughout task epochs. In our experiments, we focused on the predicted value associated with each conditioned stimulus, which is crucial for understanding how predictive stimuli produce motivated behavior (*Berridge, 2004*). Surveys of value coding in primate PFC have found individual neurons correlated with stimulus-predicted value in many subregions, with the strongest representations typically in ORB (*Roesch and Olson, 2004*; *Sallet et al., 2007*; *Kennerley et al., 2009*; *Hunt et al., 2018*). In rodents, there is also a rich literature on value signaling in ORB (*Schoenbaum et al., 2003*; *van Duuren et al., 2009*; *Sul et al., 2010*; *Stalnaker et al., 2014*; *Namboodiri et al., 2019*; *Kuwabara et al., 2020*; *Wang et al., 2020a*), but there have also been many reports of value-like signals in frontal cortical regions beyond ORB (*Otis et al., 2017*; *Allen et al., 2019*; *Wang et al., 2020a*; *Kondo and Matsuzaki, 2021*). In our present experiment, we sought to expand upon these rodent results by separating cue activity from licking, which tracks the value and may confound interpretation, by including more than two cue types, which provided a rich space to assess value coding, and by sampling from many frontal regions in the same experiment.

When considering the number of neurons responsive to cues rather than licks, our data confirmed the importance of ORB, which has more cue-responsive neurons than the motor and other prefrontal regions, but, beyond cue responsiveness, we were interested in identifying specific cue coding patterns pertaining to value. By analyzing the activity of cue-responsive neurons across all six odors predicting varying probabilities of reward, we were able to isolate neurons coding value, as well as those with value-like signals that could easily be misconstrued as value-coding in a task with fewer cues and value levels. Included in the value-like models are coding patterns that bias their activity for higher value odors without fitting our strict linear ranked value criteria; for instance, selective firing for one or two of the CS+ odors. The enrichment of these models among cue responsive neurons, even in the olfactory cortex, indicates the prevalence of value-biased coding schemes for odor-responsive neurons across brain regions. The question remains of where odor information is first shaped according to value. There have been multiple reports of some association-related modification of odor representations as early as the olfactory bulb (*Doucette et al., 2011*; *Li et al., 2015*; *Chu et al., 2016*; *Koldaeva et al., 2019*). Considering we detected value and especially value-like coding in AON, DP, and TTd, perhaps these regions are a crucial first step in processing and amplifying task-related input from the

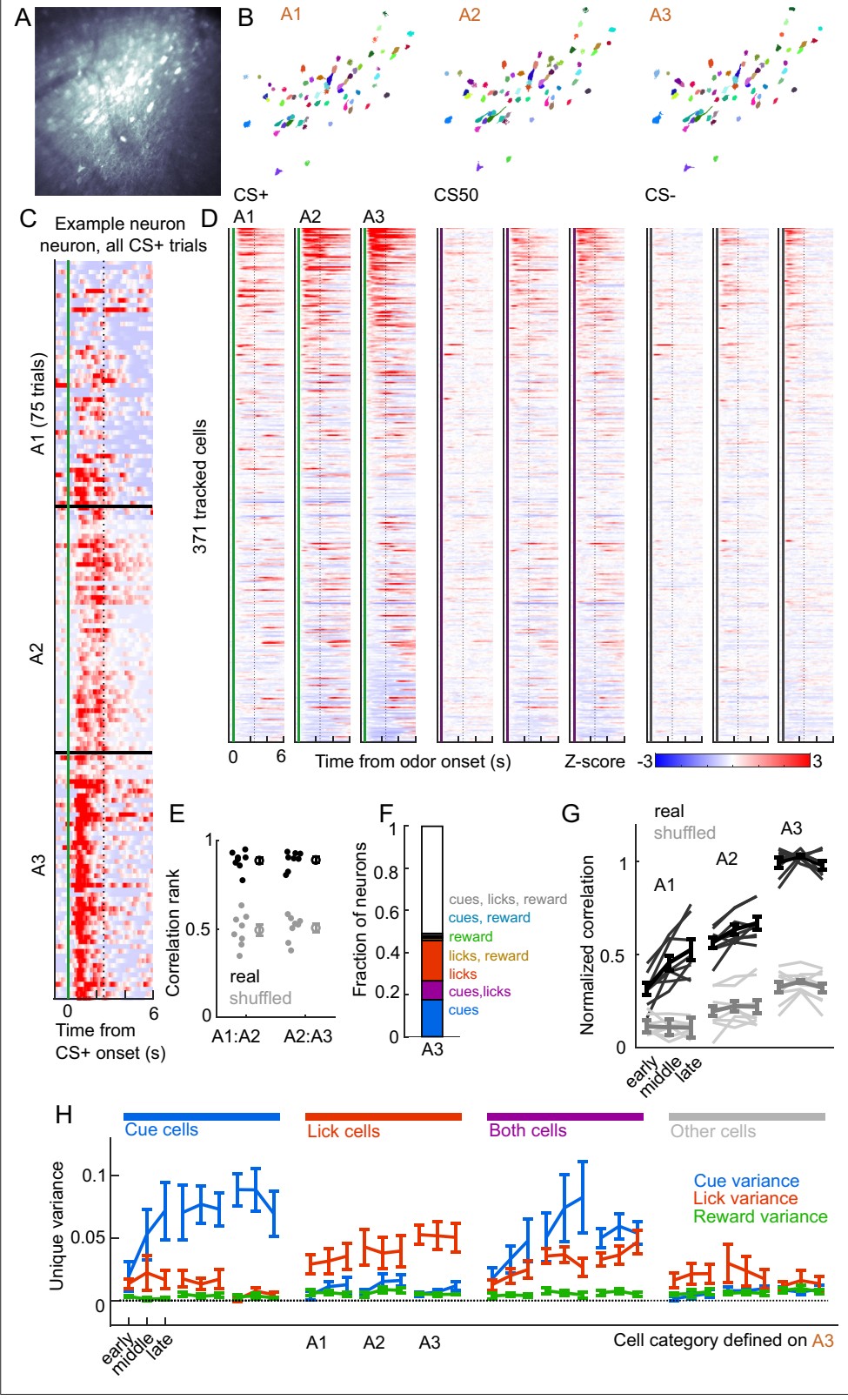

**Figure 7.** Cue and lick coding is stable across days. (**A**) Standard deviation fluorescence from example imaging plane. (**B**) Masks (randomly colored) for all tracked neurons from this imaging plane. (**C**) Deconvolved spike rate on every CS+ trial from all three sessions of odor set A for an example neuron. Vertical dashed line is reward delivery. Color axis as in (**D**). (**D**) Normalized deconvolved spike rate for all tracked neurons on all three sessions of odor

*Figure 7 continued on next page*

*Figure 7 continued*

set A. (**E**) Correlation between the activity of a given neuron in one session and its own activity in the subsequent session, quantified as a percentile out of correlations with the activity of all other neurons on the subsequent day. Plotted as the median for each subject and the mean (± SEM) across these values. Real data was more correlated than shuffled data ($p = 0.0078$ for both comparisons, Wilcoxon signed-rank test). (**F**) Fraction of tracked neurons coding cues, licks, rewards, and their combinations on day 3. (**G**) Model performance when using models from session A3 to predict the activity of individual neurons across session thirds of odor set A training, plotted as mean (± SEM) correlation between true and predicted activity across mice, normalized to the correlation between model and training data. Thin lines are individual mice. Performance was greater than shuffled data at all time points ($p < 0.002$, Bonferroni-corrected, $n = 8$ mice). Non-normalized data in *Figure 7—figure supplement 1*. (**H**) Mean (± SEM across mice) unique cue, lick, and reward variance for cells classified as coding cues, licks, both, or neither on session A3. A3 cue cells had increased cue variance in A2 ($p < 10^{-7}$, see Methods) and A1 ($p < 0.03$) relative to lick and reward variance. Same pattern for A3 lick cells in A2 ($p < 0.0001$) and A1 ($p < 0.01$).

The online version of this article includes the following figure supplement(s) for figure 7:

**Figure supplement 1.** Correlation across days in prelimbic area (PL).

olfactory bulb. Because they provide input to PFC (*Igarashi et al., 2012*; *Bhattarai et al., 2022*), they may be an important source of the cue coding we observed there.

The distribution of cue cells with linear coding of value was mostly even across regions, with slight enrichment overall in PFC compared to the motor and olfactory cortex, but no subregional differences in PFC. Importantly, cue value could be decoded from value cells in each region with similar accuracy. One consequence of a widely distributed value signal is that manipulating only one subregion would be less likely to fully disrupt value representations, which is consistent with the results of studies comparing functional manipulations across PFC (*Chudasama and Robbins, 2003*; *St Onge and Floresco, 2010*; *Dalton et al., 2016*; *Verharen et al., 2020*; *Wang et al., 2020a*). Different subregional impacts on behavior may reveal biases in how the value signal in each region contributes to reward-related behaviors, for instance during learning or expression of a reward associations (*Otis et al., 2017*; *Namboodiri et al., 2019*; *Wang et al., 2020a*). A related interpretation is that, in this task, there may be other properties that correlate with cue value, and the homogeneous value representation we observed across regions masks regional differences in tuning to these other correlated features, such as motivation (*Roesch and Olson, 2004*) and a host of related concepts, including salience, uncertainty, vigor, and arousal (*Stalnaker et al., 2015*; *Hayden and Niv, 2021*; *Zhou et al., 2021*), which can have different contributions to behavior. This interpretation is consistent with broader views that observations of 'value' signals are often misconstrued (*Zhou et al., 2021*) and that pure abstract value may not be encoded in the brain at all (*Hayden and Niv, 2021*). Although the identification of value in our task was robust to three levels of reward probability across two stimulus sets, the fact that this signal was widespread contributes to the case for revisiting the definition and interpretation of value to better understand regional specialization.

In our analysis, we uncovered a distinction between neurons encoding the overall value of cues and those with value representations that incorporated the recent reward history. Neurons with history effects were rare and most frequent in PFC. These neurons may have a more direct impact on behavioral output in this task, because the lick rate also incorporated recent reward history. Notably, the impact of reward history on these neurons was noticeable even prior to cue onset, consistent with a previously proposed mechanism for persistent value representations encoded in the baseline firing rates of PFC neurons (*Bari et al., 2019*).

## Stability of PFC codes

Previous reports have observed drifting representations in PFC across time (*Hyman et al., 2012*; *Malagon-Vina et al., 2018*), and there is compelling evidence that odor representations in piriform drift over weeks when odors are experienced infrequently (*Schoonover et al., 2021*). On the other hand, it has been shown that coding for odor association is stable in ORB and PL, and that coding for odor identity is stable in piriform (*Wang et al., 2020a*), with similar findings for auditory Pavlovian cue encoding in PL (*Otis et al., 2017*; *Grant et al., 2021*) and ORB (*Namboodiri et al., 2019*). We were able to expand upon these data in PL by identifying both cue and lick coding and showing separable, stable coding of cues and licks across days and across sets of odors trained on separate days. We

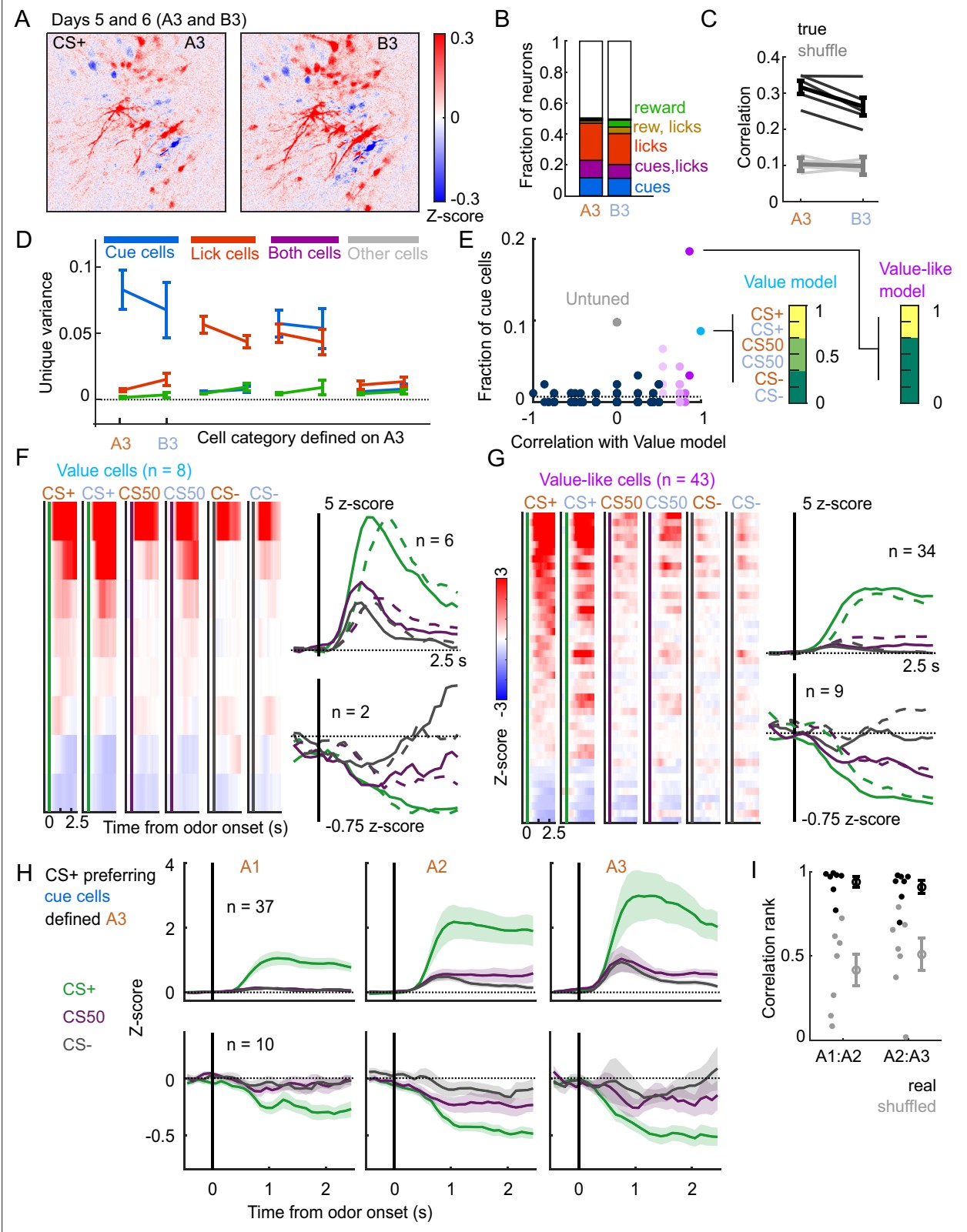

**Figure 8.** Stable cue coding across separately trained odor sets. (**A**) Normalized activity of all pixels in the imaging plane following CS+ presentation on the third day of each odor set (A3 and B3, days 5 and 6 of training). (**B**) Fraction of neurons coding for cues, licks, rewards, and their combinations in A3 and B3 (days 5 and 6). (**C**) Mean (± SEM, across mice) correlation between activity predicted by odor set A3 models and its training data (A3, cross-validated) or activity in B3, for true (black) and trial shuffled (gray) activity. Thin lines are individual mice. $F_{(1, 16)} = 3.2$, $p = 0.09$ for main effect

*Figure 8 continued on next page*

*Figure 8 continued*

of odor set, $F(1, 16) = 135$, $p < 10^{-8}$ for main effect of shuffle, $F(1, 16) = 2.2$, $p = 0.16$ for interaction, $n = 5$ mice, two-way ANOVA. (**D**) Mean (± SEM, across mice) unique cue, lick, and reward variance for cells classified as coding cues, licks, both, or neither for odor set A. For each category, odor set A unique variance preference was maintained for odor set B ($p < 0.04$) except for both cells, for which lick and reward variance were not different in odor set B ($p = 0.22$, Bonferroni-corrected, $n = 5$ mice). (**E**) Distribution of best model fits across all cue cells, with colors from *Figure 3C*. Dashed line is chance proportion when assuming even distribution. (**F**) Left: normalized activity of every value cell, sorted by mean firing 0–1.5s following odor set A CS+ onset. Right: mean normalized activity of all value cells, grouped by whether peak cue activity (0–2.5 s) was above (top) or below (bottom) baseline in held out trials. Number of neurons noted for each plot. (**G**) As in (**E**), for value-like cells. (**H**) Mean (± SEM, across neurons) activity of cue cells tracked across A1, A2, and A3 with preferential CS+ firing, defined on half of A3 trials and plotted for the other half of A3 trials and all of A1 and A2 trials. (**I**) For neurons in (**H**), correlation between a neuron's activity in one session and its own activity in the subsequent session, quantified as a percentile out of correlations with the activity of all other neurons on the subsequent day. Plotted as the median for each subject ($n = 7$ with CS+ preferring cue cells) and the mean (± SEM) across these values. Real data was more correlated than shuffled data ($p = 0.016$ A1:A2, $p = 0.031$ A2:A3, Wilcoxon signed-rank test).

were also able to detect value coding common to two stimulus sets presented on separate days, and conserved value features across the three training sessions. Notably, the model with responses only to CS+ cues best fit a larger fraction of imaged PL neurons than the ranked value model, a departure from the electrophysiology results. It would be interesting to know if this is due to a bias introduced by the calcium imaging approach, the slightly reduced CS50 licking relative to CS+ licking in the imaging cohort, or the shorter imaging experimental timeline.

The consistency in cue and lick representations we observed indicates that PL serves as a reliable source of information about cue associations and licking during reward-seeking tasks, perhaps contrasting with other representations in PFC (*Hyman et al., 2012*; *Malagon-Vina et al., 2018*). Interestingly, the presence of lick, but not cue coding at the very beginning of the first session of training suggests that lick cells in PL are not specific to the task but that cue cells are specific to the learned cue-reward associations. Future work could expand upon these findings by examining stimulus-independent within session value coding across many consecutive days.

Overall, our work emphasizes the importance of evaluating the regional specialization of neural encoding with systematic recordings in many regions using the same task. Future work will clarify whether cue value is similarly widely represented in other reward-seeking settings and whether there are regional differences in the function of the value signal.

## Materials and methods
### Subjects
Subjects ($n = 5$ for electrophysiology, $n = 8$ for calcium imaging) were male and female C57BL/6 mice single-housed on a 12 hr light/dark cycle and aged 12–28 weeks at the time of recordings. Imaging experiments were performed during the dark cycle, electrophysiology during the light cycle. Mice were given free access to food in their home cages for the duration of the experiment. Mice were water restricted for the duration of the experiments and maintained at around 85% of their baseline weight (*Guo et al., 2014a*). All experimental procedures were performed in strict accordance with protocols 4450–01 and 4461–01 approved by the Animal Care and Use Committee at the University of Washington.

### Surgical procedures
Mice were anesthetized with isoflurane (5%) and maintained under anesthesia for the duration of the surgery (1–2%). Mice received injections of carprofen (5 mg/kg) prior to incision.

#### Electrophysiology
A brief (1 hr) initial surgery was performed, as previously described (*Guo et al., 2014b*; *Steinmetz et al., 2017*; *Steinmetz et al., 2019*), to implant a steel headbar (approximately 15 × 3 × 0.5 mm, 1 g) for head fixation and a 3D-printed recording chamber exposing the skull for subsequent craniotomies. Briefly, an oval incision was made extending from the interparietal bone to the frontonasal suture, skirting the ocular area. The skin and periosteum were removed to expose the entire dorsal surface of the skull. Skull yaw, pitch, and roll were leveled, and exposed bone was texturized with a

brief application of green activator (Super-Bond C&B, Sun Medical). The incision was secured to the skull with the application of cyanoacrylate (VetBond; World Precision Instruments), and the 3D-printed recording chamber was attached to the skull with L-type radiopaque polymer (Super-Bond C&B). A thin layer of cyanoacrylate was applied to the skull inside the chamber and allowed to dry. Multiple (2-4) thin layers of UV-curing optical glue (Norland Optical Adhesives #81, Norland Products) were applied to the skull inside the chamber and cured with UV light to protect the exposed bone. The headbar was attached to the skull over the interparietal bone posterior to the chamber with Super-Bond polymer, and more polymer was applied around the headbar and chamber. Following recovery, a second brief (15–30 min) surgery was conducted to perform craniotomies for Neuropixels probe insertion. Briefly, following induction of anesthesia a small (2 × 1.5 mm (w × h)) craniotomy was made over the frontal cortex (+2.5–1 mm AP, ± 2.5–0.3 mm ML) with a handheld dental drill. The craniotomy was covered with a soft silicone gel (DOWSIL 3–4680) and the recording chamber was covered with a 3D-printed lid sealed with Kwik-Cast elastomer to protects craniotomy from dust.

## Calcium imaging

A Gradient-Refractive Index (GRIN) lens and metal headcap were implanted following previously described procedures (*Namboodiri et al., 2019*) with the following modifications. In most mice, once the dura was removed from the craniotomy, we performed two injections of 0.5 $\mu L$ of virus (1 $\mu L$ total) containing the GCaMP gene construct (AAVDJ-CamKIIa-GCaMP6s, $5.3 * 10^{12}$ viral particles/mL from UNC Vector core lot AV6364) using a glass pipette microinjector (Nanoject II) at Bregma +1.94 mm AP, 0.3, and 1.2 mm ML, –2 mm DV. Ten minutes elapsed before the microinjector withdrawal to allow the virus to diffuse away from each infusion site. Then, mice were implanted with a 1 × 4 mm GRIN lens (Inscopix) aimed at +1.94 mm AP, 0.6 mm ML, and –1.8 mm DV. A subset of mice did not receive viral injections; instead, a lens with the imaging face coated 1 $\mu L$ of the GCaMP6s virus mixed with 5% aqueous silk fibroin solution (*Jackman et al., 2018*) was implanted at the same coordinate. GCaMP expression and transients were similar in both preparations. Mice were allowed to recover for at least 5 weeks before experiments began.

## Behavioral training

Mice were headfixed during training and recording sessions using either a headring (imaging experiments) or headbar (electrophysiology experiments). After initial habituation to head fixation, mice were first trained to lick for 2.5$\mu L$ rewards of 10% sucrose solution, delivered every 8–12 s through a miniature inert liquid valve (Parker 003-0257-900). After 4–5 days of lick training, mice experienced their first odor exposure (without reward delivery). Odors were delivered for a total of 1.5 s using a 4-channel olfactometer (Aurora 206 A) with 10% odor flow rate and 800 SCCM overall flow rate of medical air. Odors were randomly assigned to sets and cue identities, counterbalanced across mice. Odors were -carvone, -limonene, alpha-pinene, butanol, benzaldehyde, and geranyl acetate (Sigma Aldrich 124931, 218367, 147524, 281549, 418099, 173495, respectively), selected because of they are of neutral valence to naive mice (*Devore et al., 2013*; *Saraiva et al., 2016*). Odors were diluted 1:10 in mineral oil and 10 µL was pipetted onto filter paper within the odor delivery vials (Thermo Fisher SS246-0040) prior to each session. Airflow was constant onto the mouse's nose throughout the session and switched from clean air to scented air for the 1.5 s duration odor delivery on each trial.

On days 1–2 of Pavlovian conditioning, mice received 50–75 trials each of three odor cues (odor set A), followed by reward on 100% (CS+), 50% (CS50), or 0% (CS−) of trials, 2.5 s following the odor onset, with 8–12 s between odor presentations. On days 3–4 mice then received training for 2 days with a second odor set (odor set B) with three new odors. For electrophysiology experiments, the odors were subsequently presented in the same sessions in six blocks of 51 trials. Odor set order alternated and was counterbalanced across days. For imaging experiments, mice received the third day of odor set A on day 5 and the third day of odor set B on day 6 of conditioning. An additional three imaging mice were only trained on one odor set.

## Electrophysiological recording and spike sorting

During recording sessions, mice were headfixed. Recordings were made using either Neuropixels 1.0 or Neuropixels 2.0 electrode arrays (*Jun et al., 2017*; *Steinmetz et al., 2021*), which have 384 selectable recording sites. Recordings were made with either 1.0 (1 shank, 960 sites), 2.1 (1 shank,

1280 sites),, or 2.4 (4 shanks, 5120 sites) probes, depending on the regions of interest. Probes were mounted to a dovetail and affixed to a steel rod held by a micromanipulator (uMP-4, Sensapex Inc). For later electrode track localization within the brain, probes were coated with a fluorescent dye (DiI, ThermoFisher Vybrant V22888) by holding 2 $\mu l$ in a droplet on the end of a micropipette and painting the probe shank. In each session, one or two probes were advanced through the silicone gel covering the craniotomy over the frontal cortex, then advanced to their final position at approximately 3 $\mu m$ s$^{-1}$. Electrodes were allowed to settle for around 15 min before starting recording. Recordings were made in internal reference mode using the 'tip' reference site, with a 30 kHz sampling rate. Recordings were repeated at different locations on each of multiple subsequent days, performing an additional craniotomy over the contralateral frontal cortex. The resulting data were automatically spike sorted with Kilosort2.5 and Kilosort3 (https://github.com/MouseLand/Kilosort; RRID:SCR_016422; *Pachitariu et al., 2023*), v2.5 and 3.0. Extracellular voltage traces were preprocessed with common-average referencing by subtracting each channel's median to remove baseline offsets, then subtracting the median across all channels at each time point to remove common electrical artifacts. Sorted units were curated using automated quality control (*Banga, et al., 2022*): exclusions were based on spike floor violations (the estimated proportion of spikes that were missed because they fell below the noise level of the recording, estimated false negative rate), and refractory period violations (the estimated proportion of spikes arising from the non-primary neuron, the estimated false positive rate due to contamination, with a 10% cutoff). Quality control accuracy was assessed by manually reviewing a subset of the data using the phy GUI (https://github.com/kwikteam/phy; *Rossant et al., 2021*). Because Kilosort2.5 and Kilosort3 use different clustering algorithms that can be advantageous for different types of recordings (stability, region, number of channels), for each session, we used units sorted with either Kilosort2.5 or Kilosort3 depending on which yielded the greatest number of high-quality units for that session. Brain regions were only included for subsequent analysis if there were recordings from at least three subjects and a total of over 100 neurons in the region. When we analyzed all of the motor cortex together, we included ALM and MOs neurons. When we analyzed all of the olfactory cortex, we included DP, TTd, AON, and other neurons in PIR, EPd, and OLF. We relabeled PIR and EPd as OLF because there were not enough neurons to analyze them as separate regions.

## Imaging and ROI extraction

During imaging sessions, mice were headfixed and positioned under the 2-photon microscope (Bruker Ultima2P Plus) using a 20 x air objective (Olympus LCPLN20XIR). A Spectra-Physics InSight X3 tuned to 920 nm was used to excite GCaMP6s through the GRIN lens. Synchronization of odor and 10% sucrose delivery, lick behavior recordings, and 2-photon recordings were achieved with custom Arduino code. After recording, raw TIF files were imported into suite2p (https://github.com/Mouse-Land/suite2p; RRID:SCR_016434; *Stringer et al., 2023*), v0.13.0. We used their registration, region-of-interest (ROI) extraction, and spike deconvolution algorithms, inputting a decay factor of $\tau = 1.3$ to reflect the dynamics of GCaMP6s, and manually reviewed putative neuron ROIs for appropriate morphology and dynamics. To find changes in activity across the entire imaging plane, found the mean pixel intensity for frames in the time of interest (2–2.5 s from CS+), subtracted the mean intensity of each pixel prior to cue onset (−2–0 s from all cues), and divided by the standard deviation for each pixel across those frames prior to cue onset.

## Histology

Animals were anesthetized with pentobarbital or isoflurane. Mice were perfused intracardially with 0.9% saline followed by 4% paraformaldehyde (PFA).

### Electrophysiology

Brains were extracted immediately following perfusion and post-fixed in 4% paraformaldehyde for 24 h. In preparation for light sheet imaging brains were cleared using organic solvents following the 3DISCO protocol (*Ertürk et al., 2012*) (https://idisco.info/), with some modification. Briefly, on day 1 brains were washed 3 X in PBS and dehydrated in a series of increasing MeOH concentrations (20%, 40%, 60%, 80%, 100%, 100%; 1 hr each) then incubated overnight for lipid extraction in 66% dichloromethane (DCM) in MeOH. On day 2 brains were washed 2 X twice in 100% MeOH for 1 hr each, then bleached overnight in 5% $H_2O_2$ in MeOH at 4 °C. On day 3 brains were washed 2 X in 100% MeOH,

then final lipid extraction was accomplished in a series of DCM incubations (3 hr in 66% DCM in MeOH, 2X 100% DCM for 15 min each) before immersion in dibenzyl ether (DBE) for refractive index matching. Brains were imaged on a light sheet microscope (LaVIsion Biotec UltraScope II) 2–7 days after clearing. Brains were immersed in DBE in the imaging well secured in the horizontal position, and illuminated by a single light sheet (100% width, 4 $\mu m$ thick) from the right. Images were collected through the 2 X objective at 1 X magnification, from the dorsal surface of the brain to the ventral surface in 10 $\mu m$ steps in 488 $nm$ (autofluorescence, 30% power) and 594 $nm$ (DiI, 2–10% power) excitation channels. The 1000 raw TIF images were compiled into a single multi-image file with 10 $\mu m$ voxels, then spatially downsampled to 25 $\mu m$ voxels for transformation to the Allen common-coordinate framework (CCF) volume (*Wang et al., 2020b*) using the Elastix algorithm (*Shamonin et al., 2013*). CCF-transformed volumes were used to generate CCF fluorescent probe tract locations (pixel coordinates along the probe tract) using Lasagna (https://github.com/SainsburyWellcomeCentre/lasagna; *Campbell et al., 2020*). Probe tract CCF pixel coordinates (origin front, top, left) were transformed to bregma coordinates (origin bregma, x==ML, y==AP, and z==DV) in preparation for final integration with electrophysiology recordings using the International Brain Lab electrophysiology GUI (Faulkner M, Ephys Atlas GUI; 2020. https://github.com/int-brain-lab/iblapps/tree/master/atlaselectrophysiology; *Faulkner, 2020*). For recording alignment, sorted spikes and RMS voltage on each channel were displayed spatially in relation to the estimated channel locations in Atlas space from the tracked probe. The recording sites were then aligned to the Atlas by manually identifying a warping such that recording sites were best fit to the electrophysiological characteristics of the brain regions (e.g. matching location of ventricles or white matter tracts with low firing activity bands). This procedure has been estimated to have a 70 μm error (*Steinmetz et al., 2019*; *Liu et al., 2021*). Individual neuron locations were determined using the recording channel brain coordinates of each unit's maximum-amplitude waveform. We additionally assigned MOs neurons to the anterolateral motor cortex (ALM) if they were within a 0.75 mm radius of 2.5 mm AP, and 1.5 mm ML (*Chen et al., 2017*).

## Calcium imaging

Following perfusion, intact heads were left in PFA for an additional week before brain extraction. Brains were then sliced on a Leica Vibratome (VT1000S) at 70 $\mu m$ before mounting and nuclear staining via Fluoroshield with DAPI (Sigma-Aldrich F6057-20ML). Slices with GRIN lens tracks were then imaged on a Zeiss Axio Imager M2 Upright Trinocular Phase Contrast Fluorescence Microscope with ApoTome. The resulting images were manually aligned to the Allen Brain Atlas to reconstruct the location of each GRIN lens.

## Neuron tracking

To identify the same neurons across imaging sessions, we used two approaches. To track neurons across the two odor sets on days 5 and 6, we concatenated the TIF files from each session and extracted ROIs simultaneously. To track neurons across training days 1–3 for a single odor set, we manually identified ROIs from the ROI masks outputted by suite2p. We linked the ROIs using a custom Python script that permitted the selection of the same ROI across the three imaging planes using OpenCV and saved the coordinates on each day. The tracking results across days 1–3 from one subject is displayed in *Figure 7B*.

## Behavioral analysis

For electrophysiology experiments, the subject was illuminated with infrared light (850 nm, CMVision IR30) and eye and face movements were monitored. The right eye was monitored with a camera (FLIR CM3-U3-13Y3M-CS) fitted with a zoom lens (Thorlabs MVL7000) and long-pass filter (Thorlabs FEL0750), recording at 70 fps. Face movements were monitored with another camera (FLIR CM3-U3-13Y3M-CS, zoom lens Thorlabs MVL16M23, long-pass filter Thorlabs FEL0750) directed at a 2 × 3 cm mirror reflecting the left side of the face, recording at 70 fps. Licks were detected from the face video by thresholding the average intensity of an ROI centered between the lips and the lick spout, calculated for every frame. Interlick intervals were thresholded at 0.083 s for a maximum lick rate of 12 licks s$^{-1}$. For calcium imaging experiments, eye and face movements were not monitored, and licks were detected with a capacitance sensor (MPR121, Adafruit Industries) connected to an Arduino board. To determine the impact of cues and previous outcomes on anticipatory licking, we fit a linear model

on all electrophysiology sessions simultaneously (and for each mouse). We predicted the number of licks 0–2.5 s from odor onset using cue identity, outcomes on the previous 10 trials, outcomes on the previous 10 of that cue type, and the total number of presentations of that cue type so far (to account for cue-specific satiety) using 'fitlm' in MATLAB. When dividing sessions into 'early' and 'late,' we used the first 60 and last 60 trials of the session. When dividing sessions into thirds for the GLM ('early,' 'middle,' 'late'), we used even splits of trials into thirds.

## PSTH creation

Peri-stimulus time histograms (PSTHs) were constructed using 0.1 s bins surrounding cue onset.

### Electrophysiology

Neuron spike times were first binned into 0.02 s bins and smoothed with a half-normal causal filter ($\sigma = 300$ ms) across 50 bins. PSTHs were then constructed in 0.1 s bins surrounding each cue onset. Each bin of the PSTH was z-scored by subtracting the mean firing rate and dividing the standard deviation across the 0.1 s bins in the 2 s before all trials. When splitting responses by polarity (above/below baseline, *Figures 2B, 3E–F and 8H*, *Figure 2—figure supplement 2B*), we used even trials to determine polarity and plotted the mean across odd trials for cross-validation.

### Calcium imaging

Frames were collected at 30 Hz with 2-frame averaging, so the fluorescence for each neuron and the estimated deconvolved spiking was collected at 15 Hz. We interpolated the smoothing filter from the electrophysiology analysis (which was calculated at 50 Hz) and applied it to the deconvolved spiking traces. We then constructed PSTHs in 0.1 s bins surrounding each cue onset and z-scored (same as electrophysiology).

### Licks

Licking PSTHS were constructed in 0.1 s bins surrounding cue onset. Each trial was then smoothed with a half-normal causal filter ($\sigma = 800$ ms). For the GLM, the lick rate was calculated across the whole session by first counting licks in either the 0.02 s (electrophysiology) or 15 Hz (imaging) bins, smoothed with a half-normal causal filter over 25 bins, and then converted to 0.1 s bins relative to each cue.

## Kernel regression

To identify coding for cues, licks, and rewards in individual neurons, we fit reduced rank kernel-based linear model (*Steinmetz et al., 2019*).

### Data preparation

The discretized firing rates $f_n(t)$ for each neuron $n$ were calculated as described above for PSTH creation. We used the activity –1–6.5 s from each cue onset on every trial for our GLM analysis.

### Predictor matrix

The model included predictor kernels for cues (CS+, CS50, and CS− for each odor set, as relevant), licks (individual licks, lick bout start, and lick rate), and reward (initiation of consummatory bout). The cue kernels were supported over the window 0–5 s relative to the stimulus onset. The lick predictor kernels were supported from –0.3–0.3 s relative to each lick, from –0.3–2 s relative to lick bout start, and lick rate was shifted from –0.4–0.6 s in 0.2 s increments from original rate. The reward kernel was supported 0–4 s relative to first lick following reward delivery. For electrophysiology experiments, the model also included six constants that identified the block number, accounting for tonic changes in firing rate across blocks. Because not all cues were present in every block, this strategy prevented the cue kernels from being used to explain the baseline changes across blocks. For each kernel to be fit we constructed a Toeplitz predictor matrix of size $T \times l$, in which $T$ is the total number of time bins and $l$ is the number of lags required for the kernel. The predictor matrix contains diagonal stripes starting each time an event occurs and 0 otherwise. The predictor matrices were horizontally concatenated to yield a global prediction matrix $\mathbf{P}$ of size $T \times L$ containing all predictor kernels. Rate vectors of all $N$ neurons were horizontally concatenated to form $\mathbf{F}$, a $T \times N$ matrix.

### Reduced-rank regression

To prevent noisy and overfit kernels we implemented reduced-rank regression (*Steinmetz et al., 2019*), which allows regularized estimation by factorizing the kernel matrix $\mathbf{K}$ into the product of a $L \times r$ matrix $\mathbf{B}$ and a $r \times N$ matrix $\mathbf{W}$, minimizing the total error: $E = \|\mathbf{F} - \mathbf{PBW}\|^2$. The $T \times r$ matrix $\mathbf{PB}$ consists of a set of ordered temporal basis functions that can be linearly combined to estimate the neuron's firing rate over the whole training set and which results in the best possible prediction from any rank $r$ matrix. To estimate each neuron's kernel functions we generated the reduced rank predictor matrix $\mathbf{PB}$ for $r = 20$, and estimated the weights $\mathbf{w_n}$ to minimize the squared error $E_n = |\mathbf{f_n} - \mathbf{PBw_n}|^2$ with elastic net regularization (using the MATLAB function 'lassoglm') with parameters $\alpha = 0.5$ and $\lambda = [0.001, 0.005, 0.01, 0.02, 0.05, 0.1, 0.2, 0.5]$, using fourfold cross-validations to determine the optimal value for $\lambda$ for each neuron. The kernel functions for neuron $n$ was then unpacked from the L-length vector obtained by multiplying the first $r = 20$ columns of $\mathbf{B}$ by $\mathbf{w_n}$ (i.e. $\mathbf{k_n} = \mathbf{B}_{1:L,1:20}\mathbf{w_n}$). Predictor unique contributions. To assess the importance of each group of kernels for predicting a neuron's activity we first fit the activity of each neuron using the full reduced-rank regression procedure, then fit a reduced model (with fourfold cross-validation), holding out the kernels of the predictor to be tested (cues, licks, or rewards). If the difference in variance explained between the full and held-out model was > 2%, and the total variance explained by the full model was > 2%, the neuron was deemed selective for those predictors (*Steinmetz et al., 2019*). We validated this cutoff by comparing our results when adjusting the cutoff from 0.5–0.5% (*Figure 2—figure supplement 3*). The pattern of results was similar regardless of the cutoff. When we refit the reduced ranks to neural activity with the onset time of each trial shuffled, the 2% cutoff was the smallest that allowed no false positive identification of any neurons uniquely coding any variable (*Figure 2—figure supplement 3B*). Using a higher cutoff led to mislabeling neurons with clear cue responses as non-coding (*Figure 2—figure supplement 3E*).

### Cue coding models

To assess cue coding schemes, we fit a new set of models focusing on a more restricted time window (−1–2.5 s from cue onset) using only cues and licks as predictors. Cue and lick neurons were identified as before, and subsequent cue characterization was performed on neurons with only a unique contribution of cues. To identify value coding among cue neurons, we fit a new kernel models with a single cue kernel that scaled according to the cue as well as six block constants (as above) with full rank. We inputted cue values as 1, 0.5, and 0 for each CS+, CS50, and CS−, respectively, ranked according to their reward probability. We fit 152 additional models with alternative configurations of cue value: all permutations of 1, 1, 0.5, 0.5, 0, 0 across the six cues, as well as all permutations of high responses (1) for 6 (we call this the 'untuned' model), 5, 4, 3, 2, or 1 cues, with other cues set to 0. Among the 153 total models, some were more similar to the ranked value model, which we quantified by correlating the six cue values of each alternative model with the ranked model. We termed all models with a correlation greater than 0.5 as 'value-like.' For each neuron, we found the model that best explained its activity. The models, their correlation with value, and the proportion of cue neurons best fit by each model are illustrated in *Figure 3—figure supplement 1*. To verify the robustness of value coding in the neurons best fit by the ranked value model, we fit each of those neurons with 1000 iterations of the cue value model with shuffled cue order to create a null distribution. The fits of the original value model exceeded the 98th percentile of the null for all value neurons.

### History model

For a more nuanced estimation of the value of the cue on each trial, we generated per trial value predictions using the lick linear model (described in section 'Behavioral analysis') with cue type, 10 previous outcomes, and 10 previous cue outcomes as predictors. These values were used to scale the height of cue kernel on each trial and were, on average, 0.05, 0.35, and 0.5 for CS−, CS50, and CS+, respectively, but varied on each trial according to the specific reward history. We compared the performance of this model to all the other cue coding models for value and value-like neurons to find neurons better explained by the history model. For neurons better fit by the history model, we also fit 1000 additional models with shuffled trial values within each cue (disrupting the trial history effect). Neurons exceeding the 95th percentile of this null distribution were deemed history neurons.

## Coding stability

In the calcium imaging experiments, we used a number of approaches to assess the stability of neural codings. First, for neurons tracked across the first three sessions of odor set A, we took the trial-averaged activity of a given neuron for CS+, CS50, and CS− trials in one session and correlated it with the same neuron's trial-averaged activity from the subsequent session. We quantified the strength of the correlation as its percentile among correlations between that neuron in the first sessions and every other neuron on the subsequent session and compared this value to shuffle control (neuron identity shuffled) (*Figure 7E* and *Figure 7—figure supplement 1A*). To assess coding stability of these neurons, we used the kernels resulting from fitting the full model on day 3 and the predictors from each session third to predict neural activity at those time points. We assessed the accuracy of the prediction by correlating it with the true activity versus the correlation with the trial-shuffled data and present this data in original form (*Figure 7—figure supplement 1B*) and normalize to model performance when predicting the (cross-validated) data from the entire training session (*Figure 7G*). This shuffle maintains the temporal dynamics of each trial but shuffles the link between predictors on a given trial and the neural activity for that trial; correlation of predictors (like licks) across trials preserves some prediction accuracy even with this shuffle. We also trained models with data from the third day of odor set A training (A3, day 5) and tested on training days A3 and B3 (days 5 and 6). To determine how unique variable contributions (cues, licks, rewards) evolved over times, we fit our kernel regression model independently to each session third (early, middle, late) of sessions 1–3 for neurons tracked across the three odor set A sessions (*Figure 7H*). To assess value coding across the third sessions of odor set A and B (A3 and B3, days 5 and 6) we fit the 153 cues coding models described in Cue coding models to the neurons imaged on separate days (*Figure 8E*), concatenating the data from each odor set and adding a constant for each day to account for day differences found the model with the best fit for each neuron. We also looked at the stability of value-like signals across the three days of odor set A training by identifying CS+− preferring cue cells using half of the trials in session A3 and plotting the activity of those neurons for the remaining A3 trials and all trials from A1 and A2 (*Figure 8H*).

## Principal component analysis

To visualize the dominant firing pattern of PL neurons (*Figure 1—figure supplement 2*), and of value and value-like cells (*Figure 4*), irrespective of direction (excitation or inhibition), we performed principal component analysis ('PCA' in MATLAB) on the concatenated PSTHs across all six cues for the neurons of interest, with each neuron's activity normalized by peak modulation so that each neuron's concatenated PSTH peaked at –1 or 1. We then plotted the score of the top components.

## Decoding cue identity

We adapted the approach in *Ottenheimer et al., 2018* to use single units as well as random selections of pseudo ensembles to decode cue identity (out of six options) (*Figure 3G*). First, we binned the activity of each neuron into 0.25 s bins spanning –0.5–2.5 s from the onset of every cue in the sessions. These bins were labeled as 1–6 corresponding to the 6 different cues. For all decoding, we randomly selected 50 trials of each cue to use (most sessions had 51 of each cue, but a few had only 50). For single unit decoding of cue identity, we used fivefold cross-validation to train a linear discriminant model ('fitcdiscr' in MATLAB) on 80% of the data and tested correct classification of the six cues on the remaining 20%. We plotted the mean ± SEM performance over time for value, value-like, and untuned neurons, and compared their performance using an ANOVA with fixed effects of neuron type and time point and random effect of the session, making pairwise comparisons with Bonferroni correction. For population decoding, we pooled the activity between 1 and 2.5 s from cue onset (a period with stable decoding in the single unit analysis) and randomly selected groups of 1, 5, 10, 25, 50, 75, 100, or 200 value, value-like, or untuned neurons from all regions. We used the same linear discriminant model (with regularization $\gamma = 1$ in 'fitcdiscr') and fivefold cross-validations. We performed 1000 selections of neurons at each level, plotted the mean and standard deviations of classification accuracy across these iterations, and made pairwise comparisons across groups by calculating the number of iterations where the second group was greater or equal to the first; we repeated this one-way test for both directions of all pairs of groups

and used a Bonferroni corrected $\alpha$. Pattern of results was unchanged when population activity was standardized with PCA. Pattern of results was also unchanged when we trained on one odor set and tested on the other.

## Decoding cue value

Data were prepared for population decoding of cue identity, but with cues labeled as 0, 0.5, or 1 for CS−, CS50, and CS+ trials, respectively. Instead of a linear discriminant model, we used a linear model (elastic net, $\alpha = 0.5$) to regress cue value onto the activity of pseudo ensembles of neurons. To balance our model, we used 50 of each cue type for training and tested on 50 held out trials for a cue never seen by the model; this setup thoroughly tested the idea that value is encoded on a linear scale and thus should be able to generalize to a new cue in same value domain. For example to predict the value of 50 CS+ in odor set B trials, we used for training 50 trials of CS+ A, 0 trials of CS+ B, 25 trials of CS50A, 25 trials of CS50B, 25 trials of CS-A, and 25 trials of CS-B, maximizing coverage of the data while maintaining a balanced design. These models produced predicted values for each cue. We plot the predicted value for CS+ and CS− cues on the left in *Figure 3H*. To convert these predictions to an accuracy score, we labeled values from –0.25–0.25 as CS−, 0.25–0.75 as CS50, and 0.75–1.25 as CS+ (values outside this range were automatically labeled incorrect). We performed this analysis on random groups of 1, 5, 10, 25, 50, 75, 100, or 200 value, value-like, or untuned neurons (*Figure 3H*), as well as random groups of five neurons (with replacement) from each region and 25 neurons (with replacement) from each region group (*Figure 4E*). We compared region decoding to decoding using a baseline window of –0.5–0 s from odor onset using neurons from each region. We performed 1000 selections of neurons at each level, plotted the mean and standard deviation of classification accuracy across these iterations, and made pairwise comparisons for cue identity.

## Cue coding dimension

To project population activity onto the coding dimensions separating CS− activity from CS+ and CS50 activity, respectively, we adapted an approach from *Li et al., 2016*. We first max normalized the odor set A PSTH activity of each neuron to prevent neurons with particularly large z-score values from dominating the dimension. We then defined coding dimensions from the even trials of odor set A. To find the 'consensus' cue-difference coding dimension across the group defined by each neuron's maximal difference across cue responses, we found the 0.5 s bin in the range 0–2.5 s from cue onset with the peak difference between CS− and CS+ activity or CS- and CS50 activity, for each neuron. This comprised a difference vector of length $N$ defining the maximum cue difference coding across the neuron group. This difference vector was then multiplied by the original z-score values of each neuron's peak difference bin to find the values of peak CS+ vs CS− coding; these values were used to transform the data onto a 0 (CS−) to 1 (CS+) relative cue coding scale. To transform population activity onto the CS− to CS+ dimension at each moment in CS+, CS50, and CS− trials, we multiplied the activity of all neurons in each 0.1 s bin of remaining odd odor set A trials (z-score) by the difference vector and used the same 0–1 scale conversion ('same odor set'). We also multiplied the activity of neurons for cues in odor set B by the difference vector ('other odor set'). We repeated the same process for CS− and CS50 activity. To find the angle between the CS+ and CS50 projections, we bootstrapped the vectors that connected baseline activity to peak activity of CS50 and CS+ along the CS-/CS+ and CS−/CS50 axes and found the angle between these vectors. To find population activity along the CS+/CS− dimension at each moment for CS50 trials of various values, we multiplied the activity (z-score) of all neurons in each 0.1 s bin of the CS50 PSTHs (grouped by value estimated from the lick linear model) by the difference vector and used the same conversion to 0–1 scale. To estimate the distribution of values along the CS+/CS− dimension for each CS50 value condition, we bootstrapped (5000 iterations, with replacement) the population projection and took the mean 1–2.5 s from odor onset. We calculated the slope of the activity on CS50 trials by linearly regressing the estimated position of the population on the CS+/CS− dimension against the value from the lick linear model used to group those trials (5000 iterations, with replacement). To compare slopes across cell groups, we generated a p-value by calculating the number of iterations where the second group was greater or equal to the first; we repeated in this one-way test for both directions of all pairs of groups and used a Bonferroni corrected $\alpha$.

## Statistics

All statistical tests were performed in MATLAB (MathWorks). To compare the fraction of neurons of a specific coding type across regions, we fit a generalized linear mixed-effects model ('fitglme' in MATLAB) with logit link function and with fixed effects of intercept and region and a random effect of the session and then found the estimated mean and 95% confidence interval for each region. For pairwise comparisons across regions, we used a specific contrast for each pair of regions ('coefTest' in MATLAB) to find the p-value that these regions differed from each other and used a Bonferroni-corrected *alpha* for significance. To compare the number of anticipatory licks on different trial types, we found the mean number of anticipatory licks for each cue in each session, and then performed a two-way ANOVA with effects of cue and subject and session as our n (*Figure 1C*). To compare the variance explained during each third of the first session, we found the mean value across neurons from each mouse and then performed a one-way ANOVA on those means with mouse as our n (*Figure 6H*). To compare day 3 model performance on true and shuffled data across each time point (*Figure 7F*), we found the mean value across neurons from each mouse at each time point and then performed a two-way ANOVA with main effects of shuffle and time point, with mouse as our n. We then calculated pairwise statistics using 'multcompare' in MATLAB with Bonferroni correction. To compare cue, lick, and reward unique variance at each time point for each cell category (determined on day 3, *Figure 7G*), we found the mean from the cells in that category in each mouse at each time point and performed a two-way ANOVA with main effects of variable and day, with mouse as our n. We then calculated pairwise statistics using 'multcompare' in MATLAB with Bonferroni correction.

## Acknowledgements

Thank you to Vijay Namboodiri and Charles Zhou for assistance with the imaging. Thank you to Noam Roth for the spike sorting quality control metrics and feedback on the decoding analysis. This work was supported by National Institutes of Health grants F32DA053714 (DJO), F31DA053706 (MMH), T32DK007247 (AJB), R37DA032750 (GDS), and P30DA048736 (GDS), a UW Center for the Neurobiology of Addiction, Pain, and Emotion 2-photon pilot project grant (DJO), a Klingenstein-Simons Fellowship in Neuroscience (NAS), and the Pew Biomedical Scholars Program (NAS).

## Additional information

### Funding

| Funder | Grant reference number | Author |
| --- | --- | --- |
| National Institute on Drug Abuse | DA053714 | David J Ottenheimer |
| National Institute on Drug Abuse | DA053706 | Madelyn M Hjort |
| National Institute of Diabetes and Digestive and Kidney Diseases | DK007247 | Anna J Bowen |
| National Institute on Drug Abuse | DA032750 | Garret D Stuber |
| National Institute on Drug Abuse | DA048736 | Garret D Stuber |

The funders had no role in study design, data collection and interpretation, or the decision to submit the work for publication.

### Author contributions

David J Ottenheimer, Conceptualization, Data curation, Software, Formal analysis, Funding acquisition, Validation, Investigation, Visualization, Methodology, Writing – original draft, Writing – review and editing; Madelyn M Hjort, Conceptualization, Data curation, Validation, Investigation, Writing – original draft, Writing – review and editing; Anna J Bowen, Data curation, Software, Validation,

Investigation, Visualization, Writing – original draft, Writing – review and editing; Nicholas A Steinmetz, Conceptualization, Resources, Software, Supervision, Funding acquisition, Methodology, Writing – original draft, Project administration, Writing – review and editing; Garret D Stuber, Conceptualization, Resources, Supervision, Funding acquisition, Methodology, Writing – original draft, Project administration, Writing – review and editing

**Author ORCIDs**
David J Ottenheimer ⓘ http://orcid.org/0000-0003-4882-1898
Madelyn M Hjort ⓘ http://orcid.org/0000-0001-9932-2349
Anna J Bowen ⓘ http://orcid.org/0000-0002-8911-2572
Nicholas A Steinmetz ⓘ http://orcid.org/0000-0001-7029-2908
Garret D Stuber ⓘ http://orcid.org/0000-0003-1730-4855

**Ethics**
All experimental procedures were performed in strict accordance with protocols 4450-01 and 4461-01 approved by the Animal Care and Use Committee at the University of Washington.

Consensus Public Review: https://doi.org/10.7554/eLife.84604.3.sa1
Author Response: https://doi.org/10.7554/eLife.84604.3.sa2

## Additional files

**Supplementary files**
• Supplementary file 1. Statistics related to *Figure 2G* and *Figure 2—figure supplement 4*. Bonferroni-corrected p-values from region contrast in generalized linear mixed-effects model.

• Supplementary file 2. Statistics related to *Figure 3*. Top: Bonferroni-corrected p-values from pairwise comparisons between the decoding accuracy of each group of neurons at each time point with their performance at baseline and with the other neuron groups at that time point. Middle, Bottom: Bonferroni-corrected p-values for pairwise comparisons of bootstrapped distributions (1000 samples) of decoding performance using increasing numbers of neurons in each group.

• Supplementary file 3. Statistics related to *Figure 4*. Top, Middle: Bonferroni-corrected p-values from region contrasts in generalized linear mixed-effects model. Bottom: Bonferroni-corrected p-values for pairwise comparisons of bootstrapped distributions (1000 samples) of decoding performance using value cells from each region.

• Supplementary file 4. Statistics related to *Figure 5*. Top: Bonferroni-corrected p-values for pairwise comparisons of bootstrapped distributions (5000 samples) of the slope of population activity of each group of neurons across CS50 trials of increasing value. Bottom: Bonferroni-corrected p-values from region contrasts in generalized linear mixed-effects model.

• Supplementary file 5. Statistics related to *Figure 2G* and *Figure 2—figure supplement 4*. Bonferroni-corrected p-values from region contrast in generalized linear mixed-effects model.

• MDAR checklist

**Data availability**
The data and code for this manuscript are publicly available at https://doi.org/10.5281/zenodo.6686927 (*Ottenheimer et al., 2022*).

The following datasets were generated:

| Author(s) | Year | Dataset title | Dataset URL | Database and Identifier |
|---|---|---|---|---|
| Ottenheimer D, Hjort M, Bowen A, Stuber G, Steinmetz N | 2022 | Electrophysiology and two-photon imaging data from an olfactory Pavlovian conditioning task | https://doi.org/10.6084/m9.figshare.21365598 | figshare, 10.6084/m9.figshare.21365598 |
| Ottenheimer DJ, Steinmetz N | 2023 | SteinmetzLab/ottenheimer-et-al-2022: Revised manuscript | https://doi.org/10.5281/zenodo.7718542 | Zenodo, 10.5281/zenodo.7718542 |

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
