## [Editor Report · eLife assessment]

This study makes **valuable** observations about the representation of "value" in the mouse brain, by using a nice task design and recording from an impressive number of brain regions. The combination of state-of-the-art imaging and electrophysiology data offer **solid** support for the authors' conclusions. The paper will be of interest to a broad audience of neuroscientists interested in reward processing in the brain.

---

## [Referee Report · Consensus Public Review]

Ottenheimer et al., present an interesting study looking at the neural representation of value in mice performing a pavlovian association task. The task is repeated in the same animals using two odor sets, allowing a distinction between odor identity coding and value coding. The authors use state-of-the-art electrophysiological techniques to record thousands of neurons from 11 frontal cortical regions to conclude that (1) licking is represented more strongly in dorsal frontal regions, (2) odor cues are represented more strongly in ventral frontal regions, (3) cue values are evenly distributed across regions. They separately perform a calcium imaging study to track coding across days and conclude that the representation of task features increments with learning and remains stable thereafter.Overall, these conclusions are interesting and well supported by the data.

The authors use reduced-rank kernel regression to characterize the 5332 recorded neurons on a cell-by-cell basis in terms of their responses to cues, licks, and reward, with a cell characterized as encoding one of these parameters if it accounts for at least 2% of the observed variance (while at first this seemed overly lenient, the authors present analyses demonstrating low false-positives at this threshold and that the results are robust to different cutoffs).

Having identified lick, reward, and cue cells, the authors next select the 24% of "cue-only" neurons and look for cells that specifically encode cue value. Because the animal's perception of stimulus value can't be measured directly, the authors created a linear model that predicts the amount of anticipatory licking in the interval between odor cue and reward presentations. The session-average-predicted lick rate by this model is used as an estimate of cue value and is used in the regression analysis that identified value cells. (Hence, the authors' definition of value is dependent on the average amount of anticipatory behavior ahead of a reward, which indicates that compared to the CS+, mice licked around 70% as much to the CS50 and 10% as much to the CS-.) The claim that this is an encoding of value is strengthened by the fact that cells show similar scaling of responses to two odor sets tested. Whereas the authors found more "lick" cells in motor regions and more "cue" cells in sensory regions, they find a consistent percentage of "value" cells (that is, cells found to be cue-only in the initial round of analysis that is subsequently found to encode anticipatory lick rate) across all 11 recorded regions, leading to their claim of a distributed code of value.

In subsequent sections, the authors expand their model of anticipatory-licking-as-value by incorporating trial and stimulus history terms into the model, allowing them to predict the anticipatory lick rate on individual trials within a session. They also use 2-photon imaging in PFC to demonstrate that neural coding of cue and lick are stable across three days of imaging, supported by two lines of evidence. First, they show that the correlation between cell responses on all periods except for the start of day 1 is more correlated with day 3 responses than expected by chance (although the correlation is low, the authors attribute this to inherent limitations of the data), and that response for a given neuron is substantially better correlated with its own activity across time than random neurons. Second, they show that cue identity is able to capture the highest unique fraction of variance (around 8%) in day 3 cue cells across three days of imaging, and similarly for lick behavior in lick cells and cue+lick in cue+lick cells. Nonetheless, their sample rasters for all imaged cells also indicate that representations are not perfectly stable, and it will be interesting to see what *does* change across the three days of imaging.

---

## [Author Response]

The following is the authors' response to the original reviews.

We thank the editor and reviewers for their careful consideration of our manuscript and very helpful feedback, which guided us in improving our manuscript. We would like to highlight three main areas of improvement in this version:

Statistical rigor: we have added more detail to justify our 2% cutoff for GLM variable coding, implemented stricter shuffling and cutoffs for value and history coding, and provided more information on the statistical significance of our pairwise comparisons across regions and groups. These go well beyond the field standard for identifying and comparing neural encoding of task features.Identification of value coding: we have implemented reviewer suggestions about kernel regression and value coding shuffles, providing even stronger evidence that value signaling among cue neurons is more prevalent than expected by chance, more prevalent than any other cue coding patterns, and present in all recorded regions. The rigor of this analysis is only possible due to our unique task design with 6 cues across two stimulus sets, and our consideration of 153 possible coding models exceeds standard practice for identifying value signals. We now implement population decoding, as well, providing additional support for a robust and widely-distributed value code.Stability of value code: we have updated our terminology to better highlight that the value signals in our imaging dataset are indeed identified across days, and we add new analysis to show conservation of value-like signals across training days.

Thanks to the reviewers’ suggestions, our manuscript now has substantially stronger support for the presence of stable and distributed cue value signaling. We address the specific points below.

**Excerpts from the Consensus Public Reviews:**
One limitation is the lack of focus on population-level dynamics from the perspective of decoding, with the analysis focusing primarily on encoding analyses within individual neurons.

To address this limitation, we now include population-level decoding analysis (new panels, Figs. 3G-H, 4E). This new analysis reveals that, although value neurons can be used to decode cue identity on par with other cue cells, value neurons are more accurate at predicting the *value* of held out cues (never seen by the model), highlighting the utility of a value signal as a way to consistently represent the value of different stimulus sets.

Moreover, we find comparable value prediction performance when using value neurons from each region (Fig. 4E), adding more support for the similarity of this signal across regions:

The authors use reduced-rank kernel regression to characterize the 5332 recorded neurons on a cell-by-cell basis in terms of their responses to cues, licks, and reward, with a cell characterized as encoding one of these parameters if it accounts for at least 2% of the observed variance. At least 50% of cells met this inclusion criterion in each recorded area. 2% feels like a lenient cutoff, and it is unclear how sensitive the results are to this cutoff, though the authors argue that this cutoff should still only allow a false positive rate of 0.02% (determined by randomly shuffling the onset time of each trial.)

We have provided more information about the 2% cutoff in a new figure, Figure 2-figure supplement 3. We reanalyzed the false positive rate and found that at a cutoff of 2% (but not 0.5% or 1%) there were no false positives (Figure 2-figure supplement 3B). Thus, we are confident that all neurons contain true task-related signals. Moreover, we found that the pattern of results remains largely unchanged as we change the cutoff over a range from 0.5% to 5%. With more stringent cutoffs, we begin to lose neurons with robust task-related responses (Figure 2-figure supplement 3E), so we continue to use the 2% cutoff in this version of the manuscript.

First, they show that the correlation between cell responses on all periods except for the start of day 1 is more correlated with day 3 responses than expected by chance (although the correlation is still quite low, for example, 0.2 on day 2).

We agree that a correlation of 0.2 does not seem like a large effect, however the variability in neuronal responses and noise level of the measurement enforce a ceiling that we can estimate by predicting data from the same session that it was trained on. We have replotted these data (new panel Fig. 7G) with the correlation normalized to the cross-validated performance on the training day’s data. This shows that the models do about half as well in session 1 and session 2 compared to session 3. The original plot is in a new supplementary figure, Figure 7-figure supplement 1B.

To further emphasize the similarity across days, we have added new panels (Fig. 7E and Figure 7-figure supplement 1A) showing that, across mice, a typical neuron was more correlated with its own activity on the subsequent day than with ~90% of the other neurons (shuffle controls, 50%).

Second, they show that cue identity is able to capture the highest unique fraction of variance (around 8%) in day 3 cue cells across three days of imaging, and similarly for lick behavior in lick cells and cue+lick in cue+lick cells. Nonetheless, their sample rasters for all imaged cells also indicate that representations are not perfectly stable, and it will be interesting to see what *does* change across the three days of imaging.

We agree that the representations are not perfectly stable and that is an interesting point of further investigation. One difference we did observe is increased cue coding across training (Figs. 6H, 7H).

Importantly, the authors do not present evidence that value itself is stably encoded across days, despite the paper's title. The more conservative in its claims in the Discussion seems more appropriate: "these results demonstrate a lack of regional specialization in value coding and the stability of cue and lick [(not value)] codes in PFC."

Due to confusing terminology on our part, the reviewers were mistaken about the timing of the experiment where we assess the stability of value coding. In the imaging sessions, odor sets were always presented on separate days. Thus, when we identify value coding in our imaged population, it is across two consecutive days with different odor sets, which is in itself evidence of a stable value code. We have updated our terminology and the text to make this clearer. We also added a new set of plots (Fig. 8H-I) showing the conservation of value-like signaling in cells we tracked across the first three sessions of odor set A, and, as above, that the correlation of these neurons across days is greater than expected by chance. These analyses lend further support to the stability of the value signal.

**Additional technical comments:**
1. The "shuffle #33" in figure 3B is confusing. The fit kernel in this shuffle shows that the "high" and "medium" responses increase above the pre-stimulus baseline. The "high" response is a combination of set 2 CS+ and set 1 CS50, both of which strongly suppressed the cell's firing over the 2.5-second window shown. Why then does the cue kernel fit these two trials predict an increase in firing rate above baseline at the 2.5-second time point? Is it a consequence of the reduced rank regression process, and if so, how? This strange-looking fit that does not well capture the response of the original cell makes me worry that the high fraction of identified "value" cells may be due to some constraint on the shuffle fits that leads them to often perform poorly.

To address this concern, we refit the value shuffle and its models using a full kernel regression model (rather than reduced ranks). It does improve the appearance of the kernel fits (updated Fig. 3B), and we now use this new approach when fitting cue coding models in the revised manuscript. The regularization inherent in reduced rank constrains the shape of the cue kernel somewhat, which contributed to the shape of the fits (although this did not negatively impact the variance explained); however, because of the importance of the shape of these alternative cue coding models to the interpretation of the analysis, we agree with the reviewers that this was worth improving. The main constraint on the value model and its shuffles, however, is that all cues must use the same template, scaled according to particular values assigned to each cue in each shuffle, which will doubtless lead to compromised (and strange-looking) fits when the shuffled values do not match the ranking of neuron’s cue activity. Critically, this constraint is applied equally to the value model and all the shuffles and would not bias the fits of any one model.

2. The "shuffle" condition when testing for value cells always assumes two high responses, two medium responses, and two low responses. This strategy doesn't account for cells that respond to only a subset of cues, as one might expect in a sparse-coding olfactory region. We suggest adding a set of shuffles where responses are split into two groups, with either 3 conditions per group or 2 in one group and 4 in the other.

We appreciate this valuable suggestion. We added all permutations of models with high responses to 6, 5, 4, 3, 2, or 1 odor cue to the analysis. We still find that the value model is the most frequent best model, displayed in new panels Fig. 3C-D and Figure 3-figure supplement 1A-B. The additional models allowed us to identify other neurons with cue activity best fit by models highly correlated with the ranked value model, which we term “value-like” neurons, including most neurons previously described as “trial-type” neurons. All 153 models and the fraction of neurons best fit by each one are depicted in Figure 3-figure supplement 1.

After implementing the changes to both the method of model fitting (full kernel regression, as noted above) and the possible alternative models, the distribution of value cells has changed slightly. All regions contain value cells, supporting our original conclusion that the value signal is distributed, but there is slight enrichment in PFC when combining these five regions together (Fig. 4A).

We have updated the conclusions of the paper accordingly:

Introduction: *“Unexpectedly, in contrast to the graded cue and lick coding across these regions, the proportion of neurons encoding cue value was more consistent across regions, with a slight enrichment in PFC but with similar value decoding performance across all regions.”*

Results: *“Interestingly, the frequency of value cells was similar across the recorded regions (Fig. 4A). Indeed, despite the regional variability in number of cue cells broadly (Fig. 2F-G), there were very few regions that statistically differed in their proportions of value cells (Fig. 4A, Figure 4-figure supplement 1). Overall, though, there were slightly more value cells across all of PFC than in motor and olfactory cortex (Figs. 4A, Figure 4-figure supplement 1). Although there were the most cue neurons in olfactory cortex, these were less likely to encode value than cue neurons in other regions (Figure 4-figure supplement 2). Value-like cells were also widespread; they were less frequent in motor cortex as a fraction of all neurons, but they were equivalently distributed in all regions as a fraction of cue neurons (Fig. 4B, Figure 4-figure supplement 1, Figure 4-figure supplement 2).”*

Discussion: *“In contrast to regional differences in the proportion of cue-responsive neurons, cue value cells were present in all regions and could be used to decode value with similar accuracy regardless of region.”* AND *“The distribution of cue cells with linear coding of value was mostly even across regions, with slight enrichment overall in PFC compared to motor and olfactory cortex, but no subregional differences in PFC. Importantly, cue value could be decoded from the value cells in all regions with similar accuracy.”*

3. On pages 11-12, the authors write "value coding is similarly represented across the regions we sampled." I feel this isn't quite what was shown: the authors have shown that all recorded regions contain a roughly comparable number of individual cells that are modulated by value, i.e. "value cells". However, the authors also showed that some recorded cells have mixed selectivity for value and other factors- it is possible that these mixed selectivity cells do vary between brain regions in their quantity or degree of value coding. Regions could potentially also vary in the dynamics of their value response, or in the trial-to-trial variability in the activity of value cells. I suggest the authors revise their original statement, for example by writing "we find a similar proportion of value-specific cells across the regions we sampled."

We thank the reviewer for carefully reviewing our claims. In addition to showing similar proportions of value cells, we also show that the value-related activity is similar (by plotting the first principal component of value and value-like cells, Fig. 4C-D) and that cue value could be decoded from the value cells in all regions with similar accuracy (new panel, Fig. 4E). We have updated the text to more accurately reflect these observations:

*“In contrast to regional differences in the proportion of cue-responsive neurons, cue value cells were present in all regions and value could be decoded from them with similar accuracy regardless of region.”*

4. We appreciate the authors' idea to introduce a history term to their value cell model but worry that the distinction between history-dependent value cells and lick/cue+lick cells in Figure 4 has gotten fuzzy. At this point, history-dependent value cells are the product of a set of steps: (1) they are identified as "cue" neurons because the cue type accounts for at least 2% of the variance, while the lick rate does not, then (2) among the cue neurons, a subset are identified as "value" neurons because their activity scales with the cue type across both odor sets, and then (3) among value neurons, the "history-dependent" value neurons show a response rate that scales with a model that predicts anticipatory licking. Our concern comes down to this: your conclusion that these cells are not licking cells hinges on the initial point that licking does not account for 2% of the observed variance in cell activity. But if you had dedicated an equal number of model parameters and selection steps to your licking model, might it still not turn out that a licking model predicts their activity as well as the history-dependent cue value model?What would bolster our confidence here would be a comparison of variance explained: if you compare the predictions of the history-dependent value-encoding cue neuron model to the predictions of a simple lick neuron model, how much better does the former predict what the cells are doing? Are all those extra parameters and selection steps really contributing to an improved description of how neurons will respond?

First, we would like to emphasize that “cue” neurons, as a population, have no discernible modulation by licks, which can be seen when comparing their activity on CS50 trials with and without reward, when licking clearly varies (Figure 2-figure supplement 2D). A new panel, Figure 5E now depicts the improvement in variance explained by the history model over a lick only model. The improvement is robust and universal. This is because even though the number of anticipatory licks per trial is used to fit the weights of our trial value model, these cue neurons have temporal dynamics that are more consistent with cue presentation than the presence of licks. We explain more below in our response to point 7.

5. The paper's title claims that the coding of cue value is both stable and distributed. While the point for value coding being distributed is well supported with analysis, the claim that cue value coding is "stable" is weaker. The authors show in Figure 6 that cue identity best accounts for unique variance among cue cells across three days of imaging, but it does not follow that cue value is similarly stable. Figure 7 shows that on day 3 of imaging, the two odor sets have similar encoding- but this analysis is only performed within day 3, not across days. Why not examine unique variance among value cells over days, as was done for a cue, lick, and both cells in Figure 6G? That seems to be an important missing piece and a logical next step. The Discussion is more conservative in its claims- "these results demonstrate a lack of regional specialization in value coding and the stability of cue and lick [(not value)] codes in PFC." But this subtlety is missing from the paper's title and introduction.

First, an important correction. “This analysis is only performed within day 3, not across days,” is a misunderstanding of our experiment brought on by our confusing terminology, which we have updated. This figure (now Figure 8) analyzes two sessions performed on consecutive days: Odor Set A day 3 (A3) and Odor Set B day 3 (B3), which constitute days 5 and 6 of our experiment (see updated panels Fig. 1B, 6A). This is why identifying value signaling across both of these sessions is justification for a stable code; by definition, it was present on two consecutive days.

A limitation of our imaging experiment prevents us from evaluating value signaling in each individual session (like we did for cues and licks). For the imaging, we only presented one odor set per session (unlike the electrophysiology, where odor sets were presented in blocks). Our method of identifying value signals relies on two odor sets, so we cannot quantify it on a per session basis in the imaging. However, to address this as best we could, we identified CS+-preferring cue cells in session A3 (odor set A day 3) and plotted them for sessions A1-A3 (Fig. 8H), which reveals a conserved value-like signal across days. We also found that the correlation of the activity of these neurons across days was higher than expected by chance (Fig. 8I).

We have edited the discussion text about coding stability, adding in more detail and caveats:

*“Previous reports have observed drifting representations in PFC across time (Hyman et al., 2012; Malagon-Vina et al., 2018), and there is compelling evidence that odor representations in piriform drift over weeks when odors are experienced infrequently (Schoonover et al., 2021). On the other hand, it has been shown that coding for odor association is stable in ORB and PL, and that coding for odor identity is stable in piriform (Wang et al., 2020a), with similar findings for auditory Pavlovian cue encoding in PL (Grant et al., 2021; Otis et al., 2017) and ORB (Namboodiri et al., 2019). We were able to expand upon these data in PL by identifying both cue and lick coding and showing separable, stable coding of cues and licks across days and across sets of odors trained on separate days. We were also able to detect value coding common to two stimulus sets presented on separate days, and conserved value features across the three training sessions. Notably, the model with responses only to CS+ cues best fit a larger fraction of imaged PL neurons than the ranked value model, a departure from the electrophysiology results. It would be interesting to know if this is due to a bias introduced by the imaging approach, the slightly reduced CS50 licking relative to CS+ licking in the imaging cohort, or the shorter imaging experimental timeline.*

*The consistency in cue and lick representations we observed indicates that PL serves as a reliable source of information about cue associations and licking during reward seeking tasks, perhaps contrasting with other representations in PFC (Hyman et al., 2012; Malagon-Vina et al., 2018). Interestingly, the presence of lick, but not cue coding at the very beginning of the first session of training suggests that lick cells in PL are not specific to the task but that cue cells are specific to the learned cue-reward associations. Future work could expand upon these findings by examining stimulus-independent value coding within session across many consecutive days.”*

6. Considering licking as the readout of value has pros and cons. Anticipatory licking may be correlated with subjective value, but certainly nonlinearly. After all, licking has a ceiling and floor (bounded rate from 0->10 Hz). Are results consistent with the objective value of the cues (which are 0, .5, 1)? Which measure better explained the data?

Thanks to this important suggestion, we tried fitting another set of models with 0, 0.5, 1 as the cue values. We found the same pattern of results. Overall, the fits were slightly better with 0, 0.5, 1, with 50.6% of potential value neurons (found with either version of the model) better fit by 0, 0.5, 1, and with mean variance explained of 0.265 with 0, 0.5, 1 (compared to 0.264 with the anticipatory lick values). Without strong evidence to choose one model over the other, we decided to use 0, 0.5, 1 because it exactly reflects reward probability, and is more objective as the reviewer notes, whereas before we relied on a noisier estimate of subjective value. We have changed the text accordingly.

7. How can a neuron encode "Cue" in a value-dependent manner and not also encode licking, given they are correlated? If the kernel window includes anticipatory licking, and anticipatory licking is by definition related to value, then how could a licking kernel not at least explain some of that neuron's variance?

The trial estimates of value from the lick linear regression are derived from typical licking patterns across all sessions and do not incorporate the particular number of licks on a given trial *or the latency of licking relative to cue onset*. Although the trial value model is predicting the number of licks on each trial, it only uses cue identity and reward history to make its prediction, so it is not tightly correlated with the stochastic licks on a given trial. And, importantly, we input the trial value as a cue kernel spanning the entire cue period, whereas lick kernels, per our definition, are restricted to a window around when licking occurs, which generously encompasses neural signals relating to both lick initiation and feedback. Licking *can* explain some of value and (history) neurons’ variance, which you can see in our new panel Fig. 5E, but it does not contribute any *unique* variance to the model. That is, with or without licks, the model performs just as well, so the activity of the neuron does not track any of the unique features of licks over cues (like whether or not the mouse licked on trial, when the mouse started licking on a given trial). Without cues, however, the model does worse, which means that the neuron’s activity is modulated by cues separately from when the mouse is licking. Thus, we can conclude the neuron encodes cues, but we have no evidence the neuron encodes licks (beyond the extent to which licks are correlated with cues). In our example fit in 5E, you can see how, although licks track value, they cannot recapitulate the temporal dynamics of this cue neuron. We added more description of this distinction in the manuscript.

8. The ordering analysis with the 89 permutations is very nice for showing across the population the "value ordered" gains are the best explanation of the neural activity. However, it doesn't tell you that any one neuron significantly encodes value, or the strength of this effect if they do. For the former, they could compare to a null distribution of shuffled order of neural vs CS data, and consider neurons for which model is better than chance ( a .05 FDR on a null distribution would be appropriate). This is important for supporting their conclusion of the fraction of neurons encoding value for each region.

In fact, with so many alternative models, the probability of a neuron being best fit by the value model but not encoding value above chance is extremely low. To confirm this, we ran the reviewer’s suggested shuffle analysis, and found that 100% of value neurons performed above the 0.05 FDR. We have added this result to the methods:

*“To verify the robustness of value coding in the neurons best fit by the ranked value model, we fit each of those neurons with 1000 iterations of the cue value model with shuffled cue order to create a null distribution. The fits of the original value model exceeded the 98th percentile of the null for all value neurons.”*

9. Similarly the 65% cutoff for trial history relative to shuffled is unusually low and therefore not convincing these neurons significantly encode the value. Usually, 95% or 99% is selected to give you a more standard significance criterion (FDR).

We have changed the cutoff to 95%. We originally selected 65% because neurons in the 65% to 95% range had clear history effects, especially at the population level, but we appreciate the importance of rigorous selection. Note this shuffle is very strict, preserving CS+, CS50, CS- ranking but shuffling within-cue fluctuations in value due to trial history. With the stricter value and history shuffling, we now observe fewer history neurons, and they are most prevalent in PFC (Fig. 5I)

10. "Regions with non-overlapping CIs were considered to have significantly different fractions of neurons of that coding type." This isn't a statistical test. Confidence intervals are not the same as significance.

We now perform Bonferroni-corrected pairwise contrasts between all regions in the generalized linear mixed effects model. We added the p-values for all the comparisons that previously relied on non-overlapping confidence intervals in supplementary tables.

**Minor comments:**
The methods are hard to read. Most of the information seems to be there but in general, paragraphs need to be read over multiple times for meaning to emerge.

We have edited for clarity, and if there are particular sections that remain unclear, we would be happy to know which ones.

Why is there a block predictor in the encoding model?

Because not every odor is present in every block, we did not want our models to use the specific cue predictors to try to account for differences in baseline activity that naturally occur across the session. Thus, each of the six blocks has its own predictor that serves as a constant that can adjust for changing baseline firing rate. Importantly, the block predictor simply marks the passage of blocks and contains no information about the odors present. We added more information about this to the methods:

*“For electrophysiology experiments, the model also included 6 constants that identified the block number, accounting for tonic changes in firing rate across blocks. Because not all cues were present in every block, this strategy prevented the cue kernels from being used to explain baseline changes across blocks.”*

Did you use an elastic net rather than a lasso? What is the alpha parameter for lasso?

We used an elastic net with alpha = 0.5. We added this information to the methods.

Figure 3F legend doesn't seem to match the figure.

Corrected.